# Manipulating attention facilitates cooperation
Claire Lugrin[1,4], Arkady Konovalov [2,4] ✉ & Christian C. Ruff [1,3] ✉

Cooperation is essential for human societies, but not all individuals cooperate to the same degree. This is typically attributed to individual motives - for example, to be prosocial or to avoid risks. Here, we investigate whether cooperative behavior can, in addition, reflect what people pay attention to and whether cooperation may therefore be influenced by manipulations that direct attention. We first analyze the attentional patterns of participants playing one-shot Prisoner's Dilemma games and find that choices indeed relate systematically to attention to specific social outcomes, as well as to individual eye movement patterns reflecting attentional strategies. To test for the causal impact of attention independently of participants' prosocial and risk attitudes, we manipulate the task display and find that cooperation is enhanced when displays facilitate attention to others' outcomes. Machine learning classifiers trained on these attentional patterns confirm that attentional strategies measured using eye-tracking can accurately predict cooperation out-of-sample. Our findings demonstrate that theories of cooperation can benefit from incorporating attention and that attentional interventions can improve cooperative outcomes.

Cooperation— working collectively to maximize a group benefit —is a hallmark of human social behavior[1,2] and is essential for the functioning of our societies[3]. Understanding why people engage in cooperation rather than behaving selfishly, and how we can therefore foster cooperation, is important for addressing pressing global challenges, such as climate change, infectious diseases, and problems associated with aging populations.

Natural selection often implies competition, but cooperation is observed in both animal and human groups[4]. Evolutionary biology offers several potential explanations for the emergence of cooperation, including kin selection, reciprocity, byproduct benefits, and group selection[5,6]. To understand and improve cooperative outcomes in human behavior, it is crucial to pin down the specific motivations and decision mechanisms underlying cooperative choices[7].

The existing literature suggests several possible motives underlying cooperation. First, cooperation may depend on the outcomes of the decisions. This is suggested by lab studies using economic games where two individuals choose to either cooperate or defect in the presence of a trade-off between individual and group outcomes. These games represent real-world situations in which an individual has the incentive to free-ride and take advantage of others - for instance, in group projects, when using public goods such as parks and transportation, or when dividing household chores. A vast behavioral literature exploring individual and group behavior in various possible designs, settings, and conditions of these games has

developed in economics, psychology, and sociology since the early 1960s[8]. In these studies, most individuals do not cooperate or defect categorically but choose systematically whether to cooperate in line with the relative size of the different payoffs[9–15].

Second, individual differences in cooperation may reflect individual beliefs and preferences. Most prominent in this respect may be social or other-regarding preferences[16–19]: People often cooperate with strangers due to their concern for other people's wellbeing[20], but could also do so due to other motives such as "warm glow" or empathy[21–23]. Non-social preferences related to specific circumstances may also influence cooperation. For example, cooperation can break down due to risk aversion: One might refuse to vaccinate due to personal fear of potential health issues. If another person then suspects that such an individual will not reciprocate and could therefore put their personal outcome (e.g., health) at risk, they might also choose not to engage in cooperation[24].

Finally, in addition to incentives and individual preferences, cooperation may also relate to attention and sampling of the information on which the choices are based. Studies using choice process data, such as gaze duration and saccades[25–32], have revealed that participants do not pay equal attention to all pieces of information but vary considerably in their eye movement patterns[33]. This suggests that individuals shift their overt attention in line with simplified information sampling strategies, often not using all available information to the same degree[34,35]. This may partially reflect

[1]Zurich Center for Neuroeconomics (ZNE), Department of Economics, University of Zurich, Zurich, Switzerland. [2]School of Psychology, University of Birmingham, Edgbaston, Birmingham, UK. [3]URPP Adaptive Brain Circuits in Development and Learning, University of Zurich, Zurich, Switzerland. [4]These authors contributed equally: Claire Lugrin, Arkady Konovalov. ✉e-mail: a.konovalov@bham.ac.uk; christian.ruff@econ.uzh.ch

goal-directed attention guided by preferences and choice strategies. For example, in social dilemmas, individuals with strong other-regarding preferences exhibit gaze patterns favoring others' payoffs[36]. Moreover, individuals who tend to play closer to equilibrium strategies—or who use a higher level of strategic sophistication—look longer at both their own and their opponent's payoffs, whereas individuals who fail to play the equilibrium strategies only sample their own payoffs[37]. More detailed eye-tracking analysis suggests that there are no distinct information sampling strategies underlying selfish and altruistic choices and that the difference in information sampling is gradual across the continuum of preference types[38].

Attention is not only guided by behavioral goals but can also be driven exogenously by characteristics of the environment and the choice-relevant stimuli[39–41], such as the positioning[42] or visual salience[43] of different choice attributes. However, previous studies of cooperative behavior have mainly examined goal-directed attention (as described above) and have by and large not considered how behavior may relate to how attention can be captured exogenously, for instance by the salience or location of the payoff information on the screen. This contrasts with the literature on choices in non-social contexts, where it has been consistently shown that value-based choices are strongly affected by exogenous attention manipulations[44–46]. For instance, studies on food choice have shown that the visual salience of an item leads to a stronger behavioral preference towards the item[47], and corresponding exogenous attention manipulations during moral decisions have shown that directing the attention of individuals to specific answers can also affect their choices[48,49].

Here, we integrate these separate perspectives and investigate how both individuals' preferences and exogenous manipulation of information presentation influence attentional information sampling subserving cooperative decision-making. To disentangle the effects of outcomes, preferences, and attentional mechanisms on cooperative decisions, we use one-shot Prisoner's Dilemma (PD) games combined with computational choice models and eye-tracking. PD games involve four distinct payoffs: reward (R) for mutual cooperation, temptation (T) for one-sided defection, punishment for mutual defection (P), and sucker's payoff (S) for one-sided cooperation. In a PD game, typically T > R > P > S, so the dominant strategy for a player is to always defect, as this strategy guarantees a higher payoff (T or P) independent of the opponent's choice. If both players defect (as they should do in Nash equilibrium if they are rational and optimal), they both receive a punishment payoff P, a worse outcome than they could have obtained if they had cooperated (R).

In our experiment, participants played 192 trials of this game with a real opponent seated in a separate room. Both players did not receive feedback on the opponent's choice after each game. To capture the effects of outcomes and distinct motives on cooperative choices in different individuals, we systematically varied the four payoff values. Specifically, we generated 96 unique PD games by varying the four payoffs between 0 and 40, with a step size of 5 Swiss francs (Fig. 1a-c). This allowed us to use computational models of choices in the game to estimate individual other-regarding preferences, risk attitudes, and subjective prior beliefs about potential cooperation by the opponent. The one-shot games we used circumvented other dynamic factors that could also influence cooperation, such as belief updating or direct reciprocity[50,51].

To evaluate the effects of information sampling on cooperation, we presented the games to participants (N = 88) as a 2 × 2 matrix and estimated their attention to different aspects of the payoff displays[52] by recording their gaze-dwelling times and saccades with a high-precision eye tracker (see "Methods"). To study exogenous attentional effects, we randomly alternated across trials between placing cooperative payoffs for the participant and their opponent in the left and right columns, fully crossed with the top and bottom rows (Fig. 1d). Moreover, we presented the payoff magnitudes using numbers included in rectangles of different sizes (Fig. 1a) that were proportional to the payoff values. These manipulations allowed us to test for the effects of both visual salience[43] and choice option positions on information sampling patterns linked to cooperative decisions. The location of options on the screen is plausibly

associated with differences in information sampling[53,54] via the habitual gaze patterns[37,42,55] associated with the left-to-right, top-to-bottom reading conventions in most Western languages[56,57].

While previous studies of social decision-making have focused on aggregate gaze patterns[37,38], we additionally studied individual temporal dynamics of gaze sequences and how they interacted with habitual viewing patterns and visual salience, leading to changes in the information sampled under different positions of payoffs on the screen. These manipulations and analyses allowed us to establish a clear causal link between attention and choices. We had no prior hypotheses on the complex relationship between the cooperative choices, gaze data, and our display manipulation, so our results are mainly exploratory. However, we used pilot data to confirm that individuals do respond to trial-by-trial variation in payoffs. Additionally, we replicated some of the behavioral analyses in a companion fMRI study (Supplementary Table 1, part b).

## Methods

### Participants

We recruited 88 healthy volunteers (48 female participants, 40 male participants, aged 18 to 35 years old, mean: 24.6, std: 3.1) from the participant pool of the Department of Economics at the University of Zurich to take part in the eye-tracking experiment. We determined the target sample size (N = 84) needed to estimate a significant correlation (significance level of 0.05, power 80%) between an individual's eye tracking measures and behavioral measures such as cooperation rate, assuming a weak relationship with a Pearson correlation coefficient of 0.3. We did not recruit participants who were economics majors or who had already taken part in similar economic exchange experiments (dictator games, trust games, prisoners dilemmas).

In parallel, another set of 88 participants recruited from the participant pool of the Department of Economics at the University of Zurich participated in a simultaneous fMRI session and acted as partners in the task. All participants were right-handed, medication-free, and with normal or corrected-to-normal eyesight (using contact lenses). All participants provided written informed consent. The study was approved by the ethics committee of the Canton of Zurich. The study was not preregistered. The data were collected between September and December 2021.

### Task

Each participant completed 192 trials of a one-shot Prisoner's Dilemma (Fig. 1a). In each trial, two participants (the main participant, playing columns, and their opponent playing rows) chose between two options labeled "A" and "B" (corresponding to cooperation or defection, with labeling counterbalanced across participants). The payoffs were determined by the choices of both players, leading to four possible outcomes. If both players cooperated, they both earned the reward payoff (R) whereas if they both defected, they both earned the punishment payoff (P < R). If one player cooperated but the other one defected, the player who defected earned the temptation payoff (T > R) while the player who cooperated received the sucker's payoff (S < P). We did not use the words "cooperation" or "defection" anywhere in the instructions or display to avoid influencing the participants' choices. The meaning of the letters (A/B as cooperation/defection) remained constant for each participant and was counterbalanced across participants. We manipulated the position of the defect or cooperate columns and rows within-subject. These positions were randomly determined for each trial and counterbalanced throughout the experiment.

At the end of the experiment, one trial was randomly selected and the decision of the second player was shown to the participants. This trial was then paid according to the choices of both participants, with one point corresponding to 1 Swiss franc (around 1 USD at the time of the experiment). The participants were also paid a 20 CHF show-up fee.

### Trial presentation and timing

We implemented the task in Matlab 2018b using the Psychophysics Toolbox 3 extension[58]. On each trial, participants were presented with a new

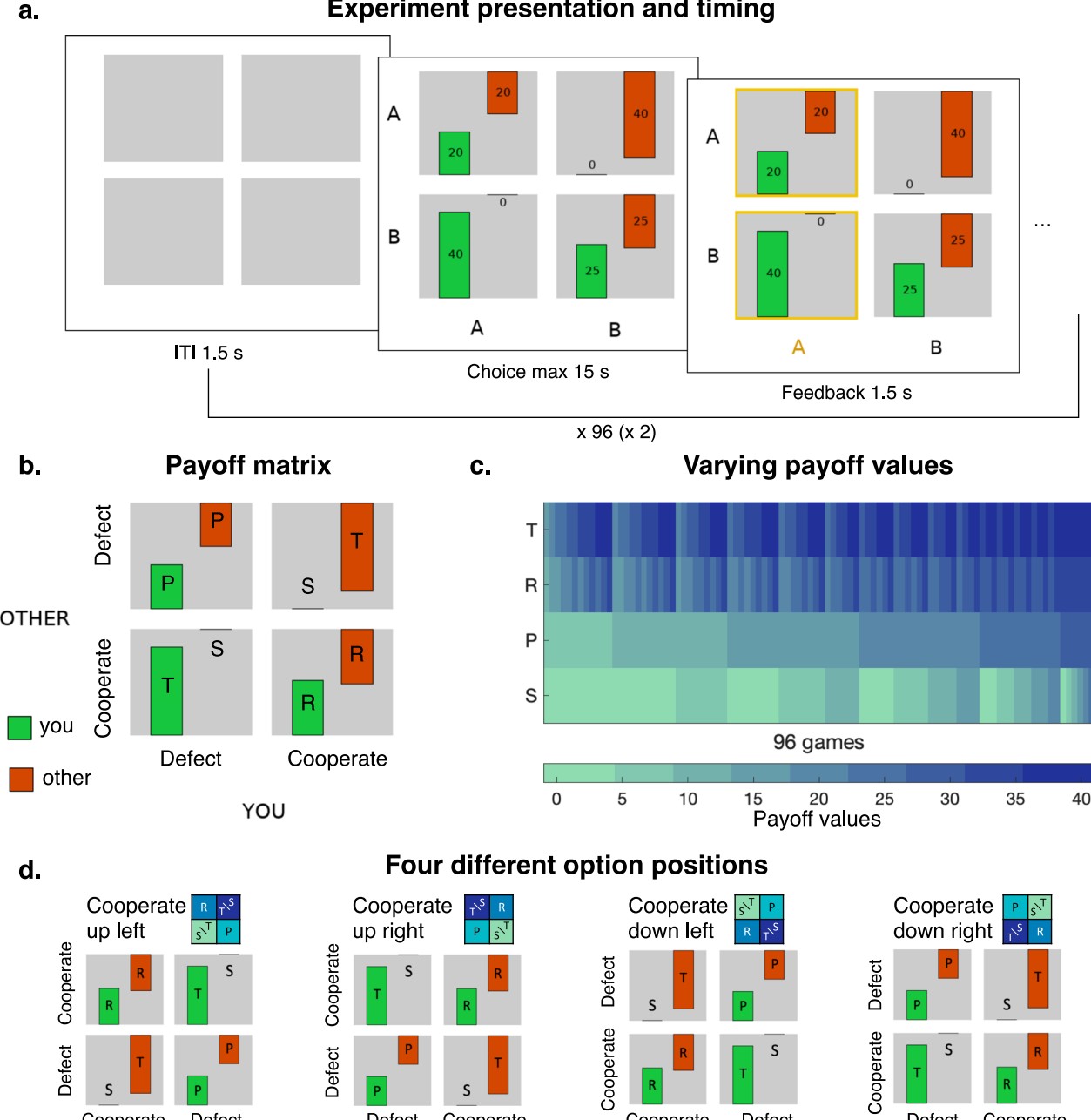

**Fig. 1 | One-shot prisoner's dilemma with varying payoffs. a** Experiment timeline. On each trial, participants chose between two columns (labeled A and B). The chosen column was indicated by yellow frames. No feedback on the other player's choice was given during the experiment. After a short inter-trial interval (ITI = 1.5 s) a new choice was presented. **b** Payoff matrix: the participant (YOU) and opponent (OTHER) chose between defecion and cooperation (labeled A and B, with labels counterbalanced across participants). If both participants cooperated (resp.

defected) they both received the reward payoff R (resp. punishment payoff P). If the participants choose different options, the participant who cooperated received the "sucker's payoff" S and the participant who defected received the temptation payoff T. **c** Values of the four payoffs in the 96 games. The values systematically varied between 0 and 40, with a step of 5, respecting the following inequalities: T > R > P > S and 2R > T + S. **d** Four possible option positions varied randomly across trials.

payoff matrix, displayed as four squares containing the potential payoffs of the participant ("you") and the second participant ("other") represented as numbers and green (for "you") and orange (for the "other") rectangles. The four squares filled the whole screen. The height of the rectangles was proportional to the payoff values (Fig. 1a, b). Participants then had a maximum of 15 s to choose either A or B by pressing the left (for the left column) or right (for the right column) arrows on the keyboard. When they chose an option, a yellow square framed the column selected for 1.5 s, and the payoffs disappeared for 1.5 s before a new trial started (Fig. 1a). Participants did not

receive any feedback on the choice of the other player until they completed all trials.

**Trial construction**

We systematically varied the four possible payoffs (reward R, temptation T, sucker's S, and punishment P) presented to the participants (Fig. 1c). Specifically, we generated all Prisoner's Dilemma games where payoffs could take values between 0 and 40 points with an increment of 5, and where the following rules were respected: T > R, P > S, R > P, T > S, and 2 R > T + S.

This procedure yielded 96 distinct games. Each participant played each game twice (96 games in random order in the first half, and the same 96 games in random order in the second half).

## Experimental session
We brought the eye-tracking and fMRI participants to the lab separately to make sure they remained anonymous. Both participants were informed about the existence of another participant playing the same game in an adjacent room, and that one of their mutual choices would be implemented at the end of the experiment. The eye-tracking participant entered the lab after the fMRI participant had started the experiment and sat in a separate room. At the end of their session, eye-tracking participants waited until the fMRI session was over. After the experiment, both participants received simultaneous feedback on one randomly selected trial.

After completing a short comprehension quiz and two practice trials, eye-tracking participants completed two blocks of 96 trials of the task each. At the end of the experiment, participants filled out a short demographic questionnaire, stating their age, sex, nationality, field of studies, religious and political orientation, monthly disposable income, family affluence, and community size.

## Eye-tracking recording
Eye tracking participants sat at a distance of about 55 cm from a 22″ screen on which we displayed the task using a resolution of 1920 × 1080 pixels. Participants placed their heads on a chinrest to reduce head movements and were instructed to hold their heads still for the duration of the block. We recorded eye motion using an infrared EyeLink 1000 Plus eye tracker system (SR Research Ltd.), sampling the gaze position at 2000 Hz. We calibrated the eye tracker before each block of 96 trials using a standard 5-point calibration routine.

## Eye-tracking processing
We used the horizontal (x) and vertical (y) positions of the participant's gaze to measure the saccades and fixations between 12 areas of interest (AOI) on the screen (Supplementary Fig. 1). Eight AOIs corresponded to the payoff of the participant ("you") and the opponent ("other") for the four possible outcomes ($R_Y$, $R_O$, $P_Y$, $P_O$, $S_Y$, $S_O$, $T_Y$, $T_O$). We recorded all fixations inside each AOI, occurring during the choice phase (between the stimulus onset and the participant's response) and lasting longer than 100 ms[59] for each trial. We recorded the sequence of fixations (AOI and location on the screen) and the duration of each fixation.

We additionally computed heatmaps of eye positions, using any x and y position (within or outside predefined AOI) recorded at each time point (Supplementary Fig. 1b), confirming that most fixations occurred within our predefined AOIs.

To analyze differences between games, or between participants, we computed trial-level metrics, using aggregates of all fixations for a trial. We computed the total time spent sampling each AOI and divided these values by the response time (duration of the trial from the stimulus onset to the response) to obtain the relative time spent sampling each AOI on each trial. We additionally analyzed the first, second, third, fourth, fifth, and last AOI sampled on each trial.

As a second proxy measuring the importance of each AOI during a trial, we computed the frequency of fixation to the different AOIs. As noted in a previous eye-tracking experiment[37], this number strongly correlated with the time spent looking at each AOI (Supplementary Fig. 2b, correlation of frequency of gaze and relative sampling time for each AOI across participants, adjusted R-squared > 0.53 for all AOIs). We therefore restricted our analysis to the relative time spent sampling each AOI.

## Computational models of cooperation
To distinguish different motives of cooperation and characterize individual differences in choice mechanisms, we built a series of computational models of cooperation. We considered three factors that can influence cooperation: other-regarding preference[16], beliefs[60], and risk[61], and included them in a single model, estimating the utility of cooperation on trial $t$ using the following equations:

$$U_t(Cooperate) = p \cdot R_t^\rho + (1 - p) \cdot ((1 - \alpha)S_t^\rho + \alpha T_t^\rho), \quad (1)$$

$$U_t(Defect) = p \cdot ((1 - \alpha)T_t^\rho + \alpha S_t^\rho) + (1 - p) \cdot P_t^\rho, \quad (2)$$

where $\alpha$ represents other-regarding preference, i.e., preference for high payoffs for the opponent ($\alpha = 1$ indicates a full consideration for the opponent's outcome and no consideration for one own's outcomes, while $\alpha = 0$ indicates no consideration for others and full consideration for self), $p$ represents the fixed individual beliefs about the opponent's action, weighing the outcomes resulting from the opponent's cooperation versus defection (thus $p = 1$ represents a sure belief that the opponent will cooperate and $p = 0$ a sure belief that the opponent will defect), and $\rho$ represents individual risk aversion using a standard power function with diminishing marginal utility.

To fit the participants' choices, we used a softmax function, computing the probability of cooperation on each trial $t$ using the difference in utility between the two options $\Delta U_t = U_t(Cooperate) - U_t(Defect)$:

$$P_t(Cooperate) = \frac{1}{1 + e^{-\beta \Delta U_t}}, \quad (3)$$

where $\beta$ is the inverse temperature ($\beta = 0$ when participants choose at random, and high $\beta$ values reflect deterministic choices).

We fitted the model to the participant's choices using standard maximum likelihood estimation (MLE), using a random grid search and optimization procedures in R (R Core Team 2017). We restricted the model fitting to participants who had enough variability in their data for the model to fit, excluding participants who defected (or cooperated) on less than two trials (22 participants), resulting in a sample size of 66 participants. The parameter recovery results are shown in Supplementary Fig. 3.

We compared the winning model to a set of alternative models, using the Bayesian Information Criterion (BIC) as a goodness-of-fit measure (see Supplementary Tables 2 and 3 for the full description of the models and comparison). Alternative models included: (1) a model that implements the risk aversion transformation after implementing other-regarding preferences, (2) a model that includes a constant bias toward cooperation or defection, (3) a model with $\alpha = 0$ (fully selfish), (4) a risk-neutral model with $\rho = 1$, (5) a model with $\alpha = 0$ (fully selfish) and a constant bias, (6) a model with a CARA risk aversion function[62], (7) a model with a CRRA risk aversion function[62], (8) a model with constant risk-seeking ($\rho = 1.1$), (9) a model with the Charness-Rabin social preference function[19], (10) a model in which the belief parameter $p$ varied with the difference in reward and punishment payoffs (R—P). We found that all these models provided a worse fit to the data than the winning model using BIC as the comparison criterion; however, BIC differences between the winning model and the risk-neutral (as well as constant risk-seeking) model were not meaningful (Supplementary Table 3).

## Statistical analysis
We analyzed the effects of payoffs, option positions, and gaze patterns using generalized mixed-effects models implemented in R using the lme4 package. We analyzed trial-by-trial choices (cooperation vs defection) and binary eye metrics (first gaze to "you" vs "other", first gaze to each payoff, etc.) using mixed-effects logistic regressions (glmer, probit link function) including random intercepts and slopes for all independent variables for each participant. We analyzed the effects of the option positions on fitted model parameters using fixed-effects linear regression and the effects of positions and payoffs on gaze duration and gaze frequency using mixed-effects linear models including random intercepts and slopes for each participant. The data were assumed to be normally distributed. All reported confidence intervals (CI) indicate 95% confidence.

### Out-of-sample predictions of cooperation

We used the trial-specific gaze patterns of each participant to obtain out-of-sample predictions of cooperation. We used random tree classifiers (fitctree implemented in MATLAB) trained on features representing the fixation of each AOI at different time points and predicting the binary outcome (cooperate vs defect) on individual trials. Specifically, we either dummy-coded the fixation of an AOI at a specific time point (first, second, third, fourth, fifth, or last fixation) or used the value of the sampled payoff on each fixation to create 32 possible features (8 AOIs × 4 fixations). We used the outcome of each trial (cooperate or defect) as labels. Using a leave-one-subject-out cross-validation procedure, we trained the classifiers using different combinations of features (either all or some fixations) on all but one participant and predicted the trial-by-trial cooperation for the left-out participant. We then averaged the predicted cooperation, either across games for each participant, or across participants for each game, and compared these out-of-sample predictions to the real participant- or game-specific cooperation rates using mean-squared-error (MSE). We additionally trained simpler classifiers, using as features the relative dwell time of each AOI on each trial, or the values of the four different payoffs on each trial, to compare the performance of classifiers trained on sequences versus summary variables.

Since our dataset was misbalanced (23% of cooperation trials and 77% of defection trials), we additionally computed the chance-level cooperation rate by shuffling the training-set labels 50 times on each cross-validation iteration and predicted the cooperation of the left-out participants using the classifiers trained on these permutations. The chance level was determined from the label permutations and estimated at around 0.2.

### Reporting summary

Further information on research design is available in the Nature Portfolio Reporting Summary linked to this article.

## Results

### Cooperation depends on payoff values, in line with other-regarding preferences

Participants cooperated on average in 23% of the trials. Importantly, cooperation rates were stable between the first and second parts of the experiment, both on the participant level (Pearson correlation $r = 0.9$, $t(64) = 15.9$, CI = [0.83 0.93]) and the game level (Pearson correlation $r = 0.72$, $t(94) = 10.05$, CI = [0.61 0.8]). The data showed individual differences in cooperation rates (from 0%, observed in 8 participants, to 100% in 1 participant). These differences may reflect individual differences in the motives underlying cooperation. To determine the influence of such motives on participants' choices, we systematically varied the payoff matrices of the games (Fig. 1c). The values of all four payoffs affected cooperation (Supplementary Table 1: mixed-effects logistic regression of payoff values on choices $p < 0.001$ for R, P, and S; $p = 0.013$ for T; see Supplementary Table 1 for CI; these effects replicated in the fMRI sample (Supplementary Table 1, part b)). Cooperation increased with larger values of the reward R and sucker's payoffs S (both of which participants can receive when choosing to cooperate) but decreased with the larger punishment payoff P (received when both participants defect) and temptation payoff T (Fig. 2a). Given that the participants played many rounds of the game, we also tested for serial effects but found no influence of the previous trial's payoffs (R, P, T, S) on the current trial's choice (mixed-effects regression of choice on previous trial R, P, T, S; $\beta_R = 0.003$, CI = [−0.02 0.02], $\beta_P = −0.003$, CI = [−0.01 0.01], $\beta_T = −0.005$, CI = [−0.02 0.01], $\beta_S = 0.005$, CI = [−0.007 0.017]).

The systematic variations of payoff values across trials allowed us to fit a computational choice model that could disentangle several factors that may influence cooperation. This model computed the utility of different choice options based on a combination of the different payoffs, weighted by their link to other-regarding preferences (preference for high payoffs for the other, $\alpha$), prior beliefs about the other participant's choice to cooperate ($p$),

and risk aversion ($\rho$, exponentially discounting the values of uncertain outcomes) (see Methods).

We compared this model to 10 alternative specifications, which showed a worse fit to the data (see Methods and Supplementary Tables 2 and 3 for details). Fitted parameters showed a high correlation between participants' cooperation rates and their other-regarding preferences (Fig. 2b, Pearson correlation between $\alpha$ and average cooperation $r(86) = 0.86$, $p < 0.001$, CI = [0.78 0.91]), but low bivariate correlations between cooperation rates and individuals' prior beliefs ($r(86) = 0.09$, $p = 0.48$, CI = [−0.16 0.32]) and risk aversion ($r(86) = 0.01$, $p = 0.94$, CI = [−0.23 0.25]), as well as a low correlation between participants' beliefs and social preferences ($r(86) = -0.18$, $p = 0.14$, CI = [-0.4 0.6]). However, in a regression model controlling for all model parameters simultaneously, we found that individual risk aversion also explained a part of the variance in cooperation rates ($R^2$ without $\rho$: 0.75, $R^2$ with $\rho$: 0.81; coefficient for $\rho$ in the full model: $\beta = −0.35$, $t(61) = 4.4$, $p < 0.001$, CI = [−0.51 −0.19]; coefficient for $\alpha$ in the full model: $\beta = 0.98$, $t(61) = 15.8$, $p < 0.001$, CI = [0.86 1.11]). There was some overlap in explained variance by the risk aversion and belief parameters: In the model that assumed risk neutrality ($\rho = 1$), the individual belief parameter became a significant predictor of cooperation ($\beta = 0.16$, $t(62) = 2.4$, $p = 0.02$, CI = [0.03 0.29]).

### Cooperation rates are linked to goal-directed attention to payoffs

Previous studies have established that goal-directed information sampling relates systematically to social preferences and individual traits[37,38] that are linked to cooperation. We confirmed these links between individual differences in cooperation and information sampling by tracking the participant's eye movements throughout the choice period until the decision. We defined several areas of interest (AOI) on the screen, placed around the 8 payoff values (Supplementary Fig. 1a: participant's own temptation, reward, punishment and sucker's payoffs $T_Y$, $R_Y$, $P_Y$, $S_Y$ and opponent's payoffs $T_O$, $R_O$, $P_O$, $S_O$) and measured the fixations on each AOI, their duration, and their temporal position during gaze sequences on each trial.

First, we focused on individual differences and investigated how each participant's aggregated gaze behavior related to differences in cooperation rates. The relative time participants spent sampling different AOIs was indeed significantly correlated with their cooperation rates (Fig. 2c). Participants who mostly sampled the other's payoffs cooperated more, whereas participants focusing on their own payoffs cooperated less (Fig. 2c, Pearson correlations of relative sampling times for each participant and AOI and participant-specific cooperation rate, $p < 0.001$ for all AOIs except for $R_Y$: $p = 0.93$, see Fig. 2c for CI). One exception was the sampling of own reward R, which was not linked to differences in cooperation. The results were similar for the trial-by-trial cooperation rates across all participants (see Supplementary Fig. 1c for details, mixed-effects logistic regression of relative sampling times on trial-by-trial cooperation: $p = 0.07$ for $P_O$, $p < 0.001$ for any other AOI, see Supplementary Table 4 for CI). Linking these results to individual preferences, we found that other-regarding preferences estimated using our utility model were correlated with the relative time participants spent sampling the other's payoffs (Supplementary Fig. 4b, Pearson's correlation: $r(86) = 0.41$, $p < 0.001$, CI = [0.19 0.59]). This confirms a link between social preferences, preferential sampling of other's payoffs, and cooperation[38].

### Cooperation is influenced by option positions

While these results confirm that cooperation is linked to individual differences in goal-directed attention, they do not imply a specific causal relation between information sampling and cooperation rates. Do preferences determine where people look, or are people's preferences influenced by what they look at? While our data cannot answer this question conclusively, we note that preferences and goals are not the only possible influence on participants' attention: Attention can also be captured exogenously by features of the environment, such as specific spatial positions that people habitually attend first, or other visual features that attract attention. We therefore

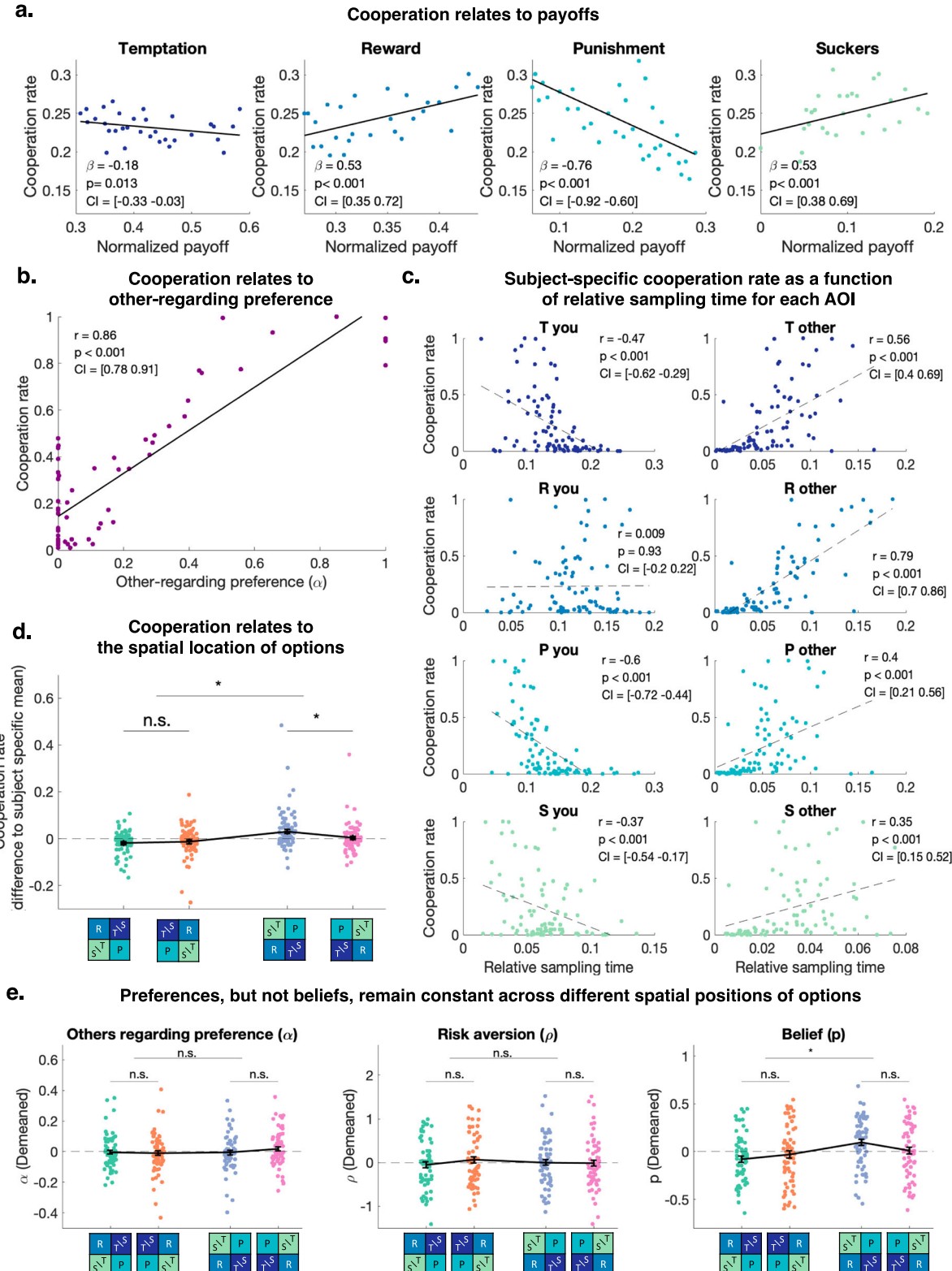

**a.** Cooperation relates to payoffs

**b.** Cooperation relates to other-regarding preference

**c.** Subject-specific cooperation rate as a function of relative sampling time for each AOI

**d.** Cooperation relates to the spatial location of options

**e.** Preferences, but not beliefs, remain constant across different spatial positions of options

capitalized on such bottom-up influences in visual sampling and exogenously manipulated attention, to test for the causal effect of this on cooperation.

To identify these bottom-up effects, we randomly permuted the positions of the options on the screen on each trial, in a within-participant and counterbalanced manner, independently from the payoff values presented. In half of the trials, the column corresponding to cooperation for the participant was placed on the left (vs right) of the screen, fully crossed with whether the cooperation option for the other was placed in the top or bottom row. This resulted in four possible combinations, which we labeled Cooperate-up-left, Cooperate-down-left, Cooperate-up-right, and Cooperate-down-right (Fig. 1d). Note that for

**Fig. 2 | Cooperation varies with payoff values, other-regarding preferences, and option positions. a** Link between payoff values and cooperation. Average cooperation rate across trials and participants for each possible value of the normalized payoffs (payoffs divided by the sum of all payoffs on a given trial). Each dot represents one normalized payoff value, the lines are fitted regressions of average cooperation rates on normalized payoffs. The beta and *p*-values correspond to fixed-effect coefficients of mixed-effects logistic regressions of payoffs on trial-by-trial cooperation (See Supplementary Table 1). *N* = 88 participants. **b** Other-regarding preference (α) measured using the choice model. Each dot represents the fitted parameter of one participant plotted against the average cooperation rate of the same participant. The line and coefficients represent the fitted regression between α and cooperation (r: correlation coefficient, p: *p*-value). *N* = 66 participants. **c** Cooperation rate plotted against average relative time spent sampling each AOI (time spent looking at the AOI divided by trial duration). Each dot represents a participant, and the lines and coefficients represent fitted regressions of cooperation rate on average relative sampling time, showing higher cooperation rates for participants who sample others' payoffs longer and lower cooperation rates for participants focus on self-payoffs. *N* = 88 participants. **d** Effect of the positions of the rows and columns on cooperation rate. Each dot represents the cooperation rate of one participant, the black error bars represent the average rate across participants and the standard error of the mean (s.e.m). The stars are *p*-values of two-tailed t-tests between the participant's cooperation rates for the different positions: *p* < 0.017 (*) or non-significant (n.s.), using a Bonferroni correction. *N* = 88 participants. **e** Fitted model parameters across the four different positions. Each dot represents the difference between the position-specific parameter and the parameter averaged across all positions of one participant, and the error bars represent the mean across participants and s.e.m. Stars represent *p*-values of two-tailed *t*-test inferior to 0.017(*) or non-significant (n.s.), applying a Bonferroni correction. *N* = 66 participants.

each spatial configuration of the payoffs, people may attend to their own payoffs or those of the other in each cell.

These changes in the positions of the options had a significant effect on cooperation. Surprisingly, the cooperation rate was higher for trials in which the cooperate option of the opponent was placed in the lower half of the screen (Fig. 2d, Cooperate-down: cooperation rate 0.25) compared to the upper half of the screen (Cooperate-up: cooperation rate 0.22, Supplementary Table 1, mixed-effects logistic regression of choices on row and columns positions: row position $p = 0.02$, $\beta = 0.16$, CI = [0.04 0.29]; this effect replicated in the fMRI sample (p = 0.01, $\beta = 0.1$, CI = [0.01 0.18], Supplementary Table 1, part b)). Including interactions between the payoff values and option positions in the regression did not alter these main effects significantly (Supplementary Table 1, column 4). Thus, cooperation decisions were causally affected by where the cooperation option was presented, regardless of the payoff values. To ensure that no serial choice effects affected these results, we tested for the interaction between the option positions and the previous trial choice. This confirmed no significant effects (mixed-effects regression of the current trial choice on the previous trial choice, interaction with row order, $p = 0.53$, $\beta = 0.11$, CI = [−0.24 0.47]).

To explain these results, we first tested whether the effects of these exogenous manipulations were linked to differences in individual preferences as captured by our model. First, we tested whether individual differences in model parameters were linked to differences in the effects of the option position manipulations, but we found no such link ($p > 0.2$ for all Pearson's correlation tests, see Supplementary Fig. 4b–d for details and CI).

Second, to test whether the option positions may alter participants' preferences, we fitted the choice model for each row and column position separately. We found that the participants' other-regarding preferences and risk aversion were not significantly different across positions (Fig. 2e; linear regression α and ρ parameters on option positions: $p > 0.5$ for all positions, see Supplementary Table 5 for CI). However, the parameter quantifying the participant's belief about the probability of cooperative choice by the opponent was significantly different between the two row positions (Supplementary Table 5, effect of row position on belief parameter: $\beta = 0.11$, $p = 0.002$, CI = [0.03 0.19]). This suggests that placing the opponent's cooperative options in the lower part of the screen might lead participants to expect their opponent to cooperate with a higher probability (across all trials, independent of the payoff variation).

**Option positions influence cooperation through changes in gaze sequences**

Changing the option positions on the screen may affect information sampling processes through interactions with habitual gaze patterns such as reading top-to-bottom and left-to-right. A possible explanation is that these positions affect the information sampled during the first saccade of each trial. Indeed, the histogram of gaze location during the first 300 ms (Fig. 3a) and the frequency of sampling different screen locations during the first fixation (Fig. 3b) showed that participants most frequently looked at the top compared to the bottom row (68% of the trials, paired two-tailed *t*-test of top

vs bottom row frequency, $p < 0.001$, $t(87) = 8.43$, CI = [27.9 45.2]) and sampled more frequently the payoffs located on the left vs right column (59% of trials; paired two-tailed *t*-test of left vs right column frequency, $p < 0.001$, $t(87) = 5.53$, CI = [12.2 25.9]). Thus, participants started by sampling the top row, but they cooperated more when cooperation was placed on the bottom. To identify what may underlie this counterintuitive effect, we first examined other features that may drive attention.

In addition to the locations of the payoffs, their denominations and values may also have an effect on exogenous capture of attention - in particular, because self versus other payoffs had specific spatial arrangements within each cell, and large payoff values corresponded to large bars in the display. In line with this, participants indeed often sampled their own payoffs (Fig. 3c: 72% of the trials) and payoffs of high values first ($T_Y$ and $R_Y$ sampled on 25% and 20% of the first fixation, the share of trials where each location is sampled increased with payoff values: see Supplementary Fig. 5a for illustration and Supplementary Table 6 for statistics).

Consistent with this assumption, we found that option positions differentially affected whether people sampled the AOIs corresponding to their own payoffs (often associated with defection, Fig. 2c) versus those associated with the others' payoffs (associated with cooperation). Own payoffs were sampled more often during the first fixation when they were positioned in the top row (Fig. 3d, mixed-effects regression of AOI locations on proportions of trials on which the AOI is sampled first, main effect of Up location, $p < 0.001$ for all own AOIs, $\beta > 0.10$, see Supplementary Table 7 for statistics). First fixations on the other's payoffs, by contrast, were mainly determined by the position of the columns, with participants looking at the other's payoffs more frequently when these were positioned on the left (Fig. 3d, mixed-effects regression of AOI locations on proportions of trials on which the AOI is sampled first: the main effect of AOI being located on the left, $p < 0.001$ for all AOIs centered on the other's payoffs, see Supplementary Table 7 for statistics).

In summary, the positioning of the options strongly affected the first saccade, with participants looking preferentially at the upper options for their own payoffs, and the left options for the other's payoff. The share of trials starting with the different AOIs for each individual correlated with their overall cooperation rate (Supplementary Fig. 5b, correlations $p < 0.01$ for all AOIs except $R_Y$, $p = 0.09$, CIs reported on the figure), showing that individual differences in first fixations were linked to individual differences in cooperation rates. When the 'Mutual cooperation' option was displayed on the bottom-left part of the screen, participants frequently started sampling the temptation payoff T of their opponent (22% of trials, indicated by an arrow in Fig. 3d). This condition was also associated with the highest cooperation rate, even though participants first looked at the worst possible outcome for themselves.

**Cooperation rates are predicted by specific gaze sequences**

Previous research has shown that not only the first saccade, but more complex gaze strategies such as scan paths, or gaze sequences, may be linked to differences in decision-making[37,55]. This suggests that exogenous manipulations of option positions as employed here may influence cooperation not only via their effects on first saccades but also by triggering more

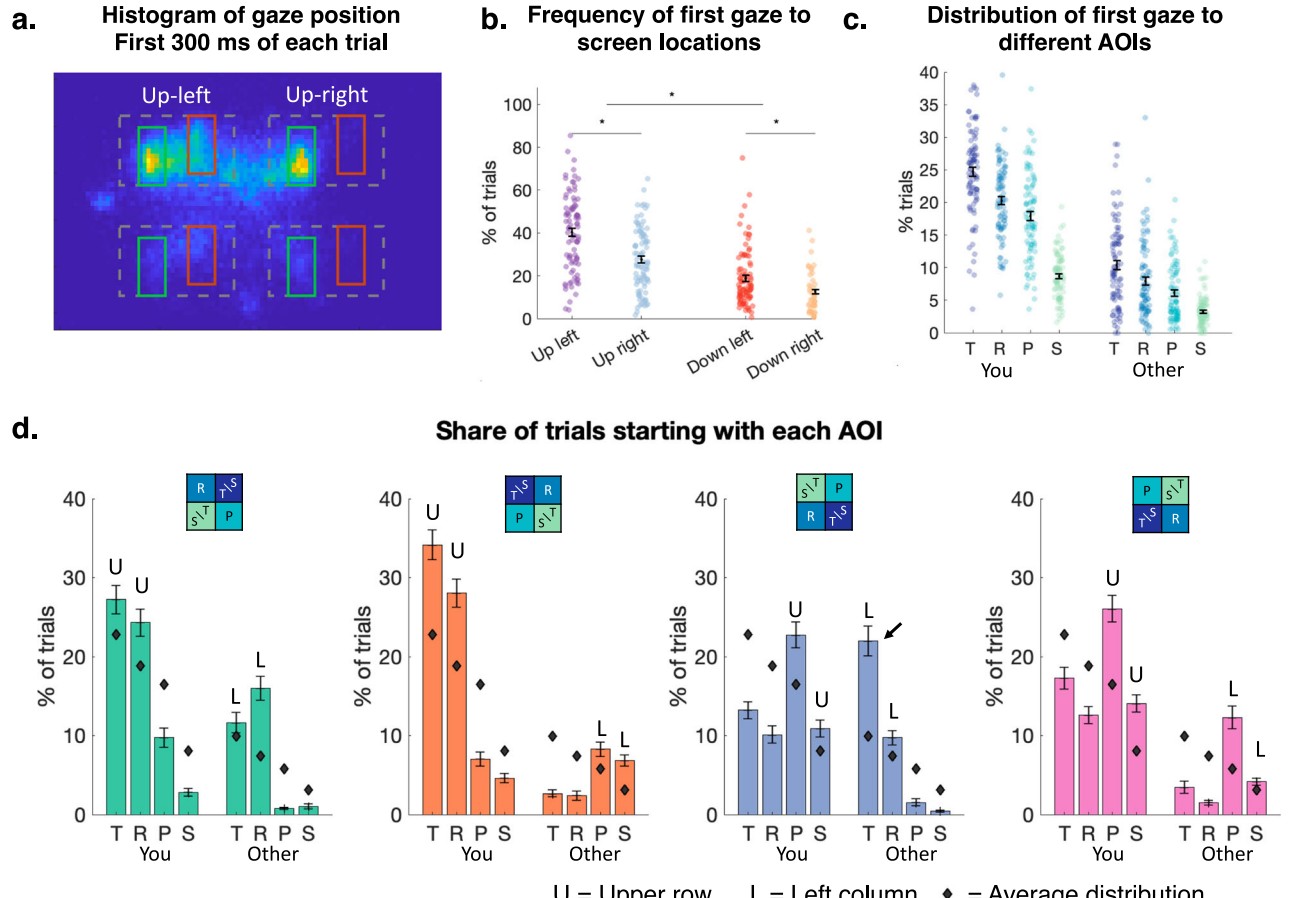

**Fig. 3 | Option positions affect the first AOI that participants sample on each trial.** a Heatmap of gaze position on the screen during the first 300 ms of each trial, across all participants and trials **b** Frequency of the first gaze being directed to four main screen locations, each dot represents the average across all trials for one participant. The error bars represent the average across participants and s.e.m. **c** Distribution of the first gaze being directed to each AOI. Each dot represents one participant. The error bars represent the average across participants and s.e.m.

**d** Share of trials starting with fixation to each AOI in the four different option positions. The colored bars represent the histogram of first fixations to each AOI for each position, averaged across participants, and the error bars represent s.e.m. across participants. The black diamonds represent the average frequency across all options positions (Fig. 3c). The arrow points to a pattern in the Mutual Cooperation down left position, where the temptation payoff T of the opponent is frequently sampled first. All panels: *N* = 88 participants.

complex extended gaze sequences. To test for these links between gaze sequences and cooperation, we extracted the temporal sequence of fixations occurring on each trial.

We first analyzed the effects of the first three fixations for each trial (a larger number was not computationally tractable for some of the analyses). This showed that cooperation was related to the temporal order of payoff sampling, with the effects of sampling a given payoff varying strongly across the different temporal positions within a gaze sequence (Fig. 4, Supplementary Table 8). For example, while sampling the payoffs received for defecting ($T_Y$ and $P_Y$) was generally associated with defection, and sampling reward payoffs R was generally associated with high cooperation (mixed-effects regression of dummies representing fixations on each AOI during the first, second, or last fixation on trial-by-trial cooperation: $p < 0.05$ for $R_Y$ and, $R_O$ first or second, $T_Y$ first, second and last, $P_Y$ last, see Supplementary Table 8 for statistics), the effects varied strongly with the timepoints at which these payoffs were sampled. Sampling the other's reward second rather than first led to an increase in cooperation rate from 12% to 18%, compared to trials during which this AOI was never sampled (Fig. 4a). Computing the most frequent gaze sequences across all trials and participants, and the cooperation rates associated with these sequences confirmed that sequences including the other's reward (R) as the second fixation had the highest cooperation rate (39%, see Supplementary Fig. 7 for frequent sequences details). These results ssuggest that beyond the first fixation, the payoffs sampled during later time points may be key for explaining cooperation.

To confirm the relevance of gaze sequences for explaining cooperative outcomes, we used machine learning classification (random tree, see "Methods" for details) to predict the out-of-sample cooperation rates across participants and games, using gaze sequences and payoff information (Fig. 4b and c).

We found that the best predictions were obtained when using the first four and the last fixations. These classifiers provided better predictions than simply using the relative dwell time, payoff values, or gazes on the first and/or last fixations (Fig. 4b), but also outperformed classifiers trained using the first 5 and last fixation. This shows that at least the first four fixations contribute to cooperative outcomes and that restricting the analysis to one or two fixations, or overall dwell time, is not enough to fully capture how behavior relates to attentional information sampling.

Machine learning predictions strongly correlated with the true cooperation rates (Fig. 4c, participant-specific predicted vs measured cooperation rate $r(86) = 0.78$, $p < 0.001$, CI = [0.68 0.85]; game-specific predicted versus measured cooperation rate $r(86) = 0.72$, $p < 0.001$, CI = [0.61 0.8]). Thus, not only the first fixations but sequential information sampling up to at least the fourth fixation was strongly linked to cooperative decisions.

**Gaze sequences are affected by spatial option presentation**
The results until now show that cooperation rates are affected not just by first fixations but also by subsequent saccades. This suggests that the spatial arrangement of the different options can influence not just where people

**a. Change in cooperation rate when each AOI us sampled on fixation 1, 2, 3, or last versus never sampled**

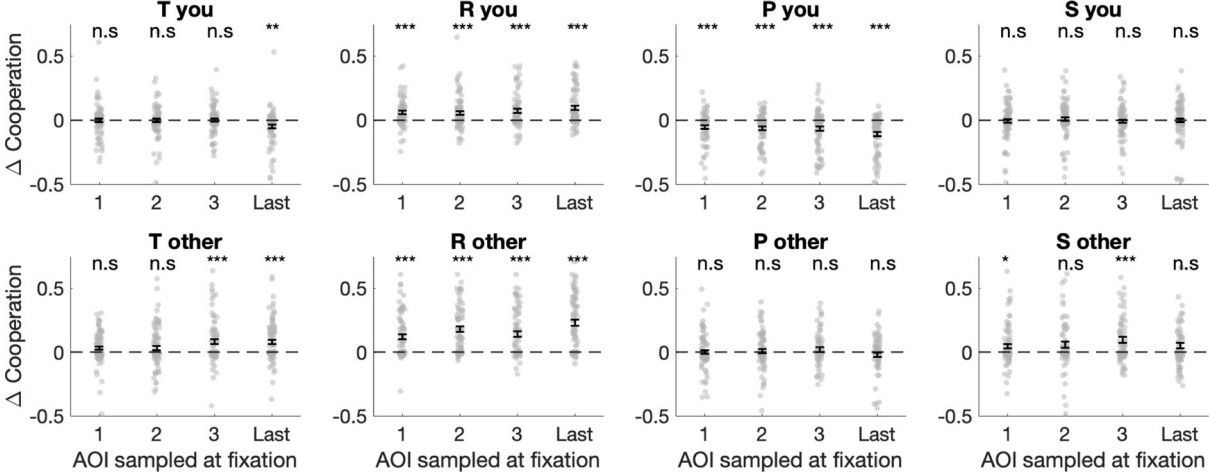

**b. Quality of fit of classifiers trained using different fixations and values**

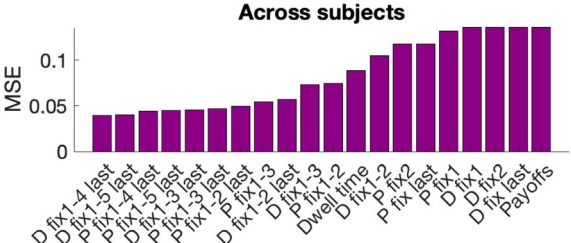
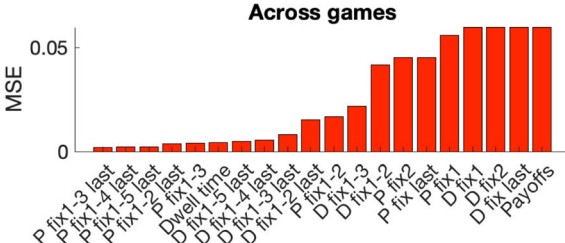

**c. Measured versus out-of-sample predicted cooperation for the winning model**

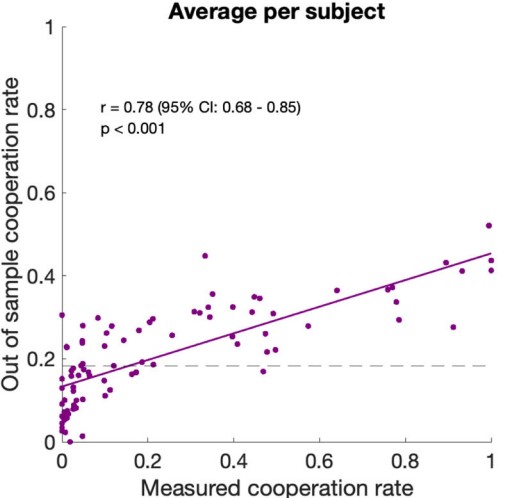
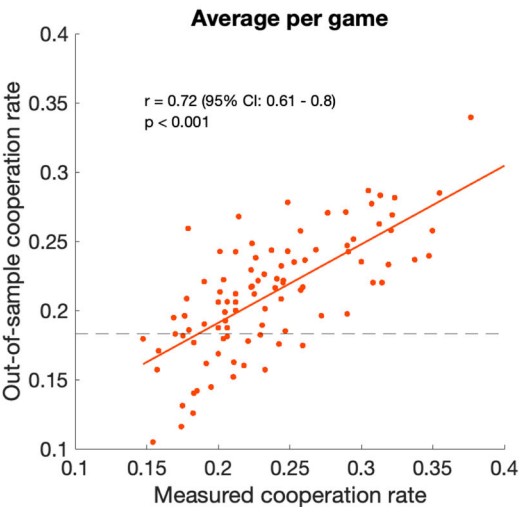

**Fig. 4 | Gaze sequences predict out-of-sample trial-by-trial cooperation. a** Gaze sequence: Differences in cooperation rates between trials in which an AOI is never looked at and trials during which it is sampled during the first, second, third or last fixation. Each dot represents the difference in cooperation rate for one participant, the error bars represent the average across participants and the s.e.m. Stars represent significant two-tailed paired t-tests ($p < 0.00025$ (***), $p < 0.0025$ (**), $p < 0.0125$ (*), using a Bonferroni correction) between cooperation rates for trials on which the AOI is never looked at and trials on which it is sampled at the specific time-point. **b** Quality of fit (mean square error) of different classifiers predicting the outcome (cooperate vs defect) on each trial using gaze patterns. The quality criterion (mean squared error) is computed using the average across trials for each participant (left) or the average across participants for each trial (right) and used to compare models capturing data the best. Models using the first four and last fixation combined with payoff values provide the best out-of-sample predictions. **c** Out-of-sample predictions of the winning classifier (payoffs; fixations 1 to 4 and last). The predictions are averaged across trials for each participant (left) or across participants for each trial (right) and plotted against the average of the true labels. The models are fitted using a leave-one-participant-out procedure, where data from all but one participant are used for training the classifier. The trained classifier is used to predict the cooperation of the left-out participant on each trial (out-of-sample predictions), which are displayed here, showing that the first four and last fixation combined with payoff values are good predictors of cooperation. The dotted lines represent chance levels, computed with permutation tests. All panels: $N = 88$ participants, except for (**c**), right $N = 96$ games across 88 participants.

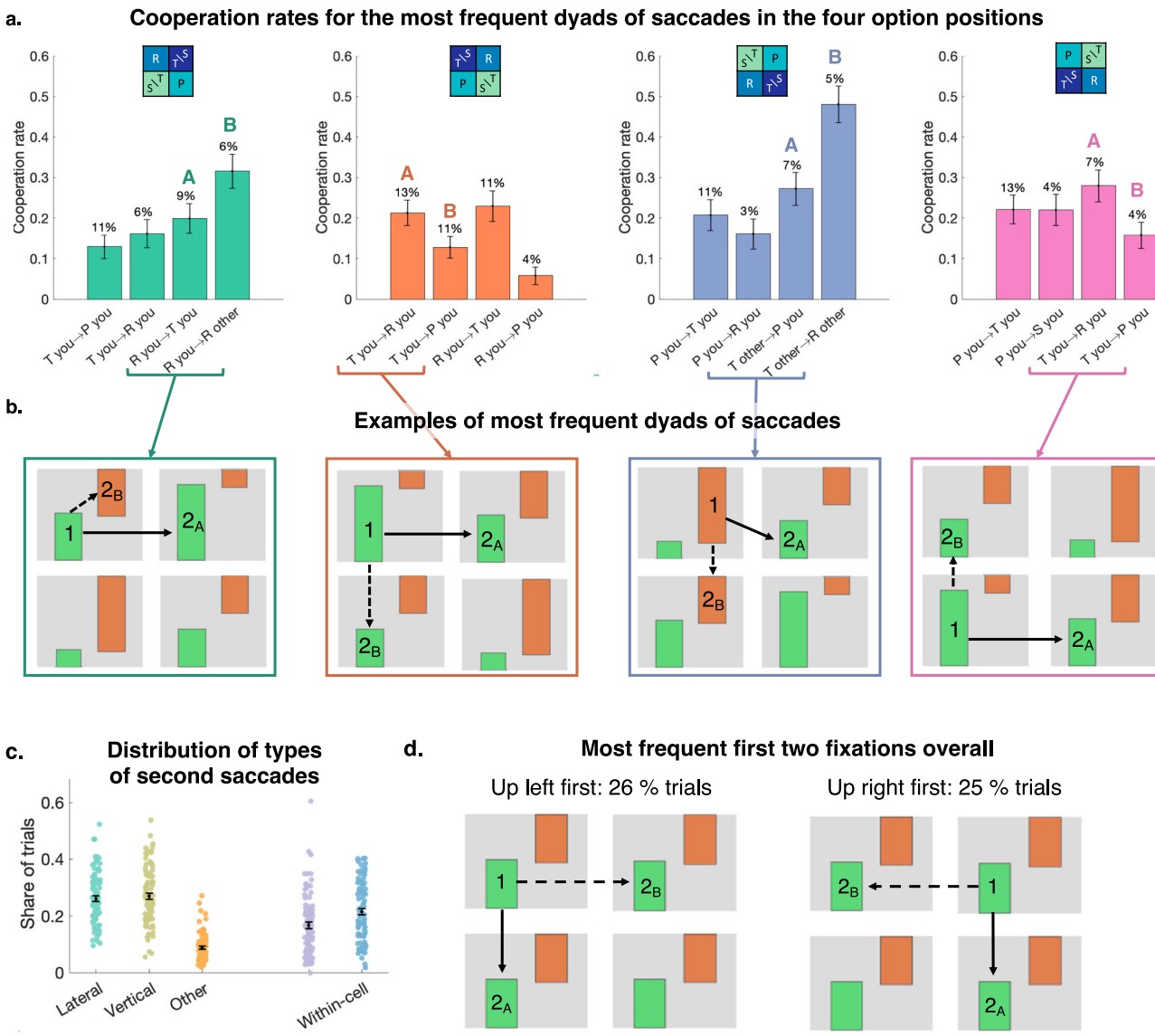

**Fig. 5 | Different option positions elicit different gaze sequences that in turn relate to different cooperation rates. a** Cooperation rate for the four most frequent gaze sequences in each position. The two most frequent first fixation and subsequent second fixation are represented (See Supplementary Fig. 7 for a more detailed description of the gaze sequences), and the cooperation rate for each participant when following these sequences is computed. The bars represent the average cooperation across participants and error bars represent the s.e.m. The percentage displayed on top of each bar is the share of all trials in the specific option position that follows the represented sequence. The letters A and B represent special gaze paths illustrated in (**b**). **b** Example of frequent gaze sequences in the different positions, where the first fixation (1) is identical, but the second gaze is different (2A and 2B), resulting in strong differences in cooperation rates (displayed in (**a**)). **c** Frequency of the second fixation being within-participant (First and second gaze to the same participant: lateral, vertical, or diagonal) or between participants (within-cell or diagonal). Each dot represents the frequency for one participant, the error bars represent mean and s.e.m. across participants. **d** Most frequent gaze patterns: on 26% of the trials, participants start looking up left at their own payoff, on 25%, they start looking up right at their own payoff. The next most frequent saccades are within-participant, lateral or towards the bottom row (see Supplementary Fig. 6 for more details). All panels: N = 88 participants.

look first, but the full sampling sequence. To test for these effects, we first computed the four most frequent fixation dyads for each option position and the cooperation rate for trials starting with these sequences (averaged across participants, Fig. 5a). This revealed that trials starting with the same fixations, but leading to different second fixations, had different cooperation rates. Figure 5b illustrates two frequent fixation dyads for each option position that lead to different cooperation rates. The Cooperation-down-left condition provides the most striking example, where a first fixation on the other's temptation T (located in the upper left cell) followed by a vertical saccade to the other's reward leads to a cooperation rate of 48%, compared to 27% when own punishment is fixated second. Exogenously driving participants' attention to a high payoff for others (associated with defection)

located in the top-left part of the screen therefore can lead to high cooperation, but mainly when participants follow a specific sampling sequence after the initial gaze.

More generally, Fig. 5a shows that for any option position, gaze dyads in which the second fixation is to the reward payoff had high cooperation rates, compared to dyads in which other payoffs are sampled second. Mixed-effects logistic regression of fixation dummies on cooperation for the four different positions (Supplementary Table 9, Supplementary Fig. 7) confirmed the strong effect of looking at the other's reward second on cooperation (positive coefficients, $p < 0.02$ for all positions except Cooperation-up-left, see Supplementary Table 9 for statistics). These examples illustrate that cooperation rates are indeed determined by different sampling

sequences – and not uniquely the first gaze—that occurs more or less frequently depending on the option positions (Fig. 5a, see differences in the percentage of sequences starting with each dyad).

Such specific sequences may reflect a combination of preferences for looking at specific payoffs—specifically others' payoffs for participants with strong other-regarding preferences (Supplementary Fig. 4b)—and habitual gaze patterns[37,42,55], affecting the frequency of sampling specific types of information not only during the first but also the following fixations. Indeed, a first Up-left gaze is frequently followed by a vertical or lateral saccade (Fig. 5c, d) that is linked to the left-to-right top-to-bottom writing and reading conventions[56,57] that are the norm for most participants of our sample (predominantly young adults of European background). Our results show that such a sampling pattern, combined with the nature of the payoffs located in the upper left area, can lead participants to sample different AOIs and thereby influence their cooperation rate.

### Changes in gaze sequences drive participants' attention to other's payoffs

In this last analysis, we focus on the differences in the type of information that is sampled in the different option positions, as a result of the changes in gaze sequences discussed before.

First, the different option positions may attract the participants' attention toward the payoffs of the opponent within a specific cell and lead them to increase their consideration of the other's outcomes in their decisions. We indeed found that when the cooperate option was placed on the left compared to the right, participants sampled others' payoffs longer (Fig. 6a, two-tailed paired t-test of differences in relative time spent sampling other's AOIs between column positions $p < 0.001$, $t(87) = 5.83$, CI = [0.02 0.04]) and also looked at these payoffs more frequently at the beginning of a trial ($p < 0.001$, $t(87) = 12.7$, CI = [0.10 0.13] see Supplementary Fig. 8 for an analysis of the effects of the first fixations to other's AOIs on cooperation that replicates the analysis presented here specifically for the first fixation of the trial). The more time participants spent sampling the other's payoffs, the higher their cooperation rates for both column positions (Fig. 6b, left panel, correlation between the relative time spent sampling the other's payoffs and cooperation rates $p < 0.001$; Cooperate left: $r(86) = 0.35$, $p < 0.001$, CI = [0.15 0.52]; Cooperate right: $r(86) = 0.37$, $p < 0.001$, CI = [0.17 0.54]).

Participants who had a stronger difference in time spent looking at the other's payoff between the two column positions also had a slightly larger increase in their cooperation rate for the Cooperate-left position (Fig. 6c, left panel, correlation between differences in time looking at other's payoff and cooperation rates between two column positions, $r(86) = 0.27$, $p = 0.011$, CI = [0.06 0.45]). This suggests that individual cooperation was influenced by the position of the columns, via participant-specific attention attraction to the payoffs of others.

Second, our finding of high cooperation rates for specific option positions may also relate to the preferential sampling of the outcomes associated with cooperation. Participants indeed sampled the cooperate row more when it was placed on the top (Fig. 6a, two-tailed paired t-test of differences in relative time spent sampling the cooperate row between row positions, $p < 0.001$, $t(87) = 6.39$, CI = [0.14 0.27]), but they cooperated more often when this row was placed at the bottom (Fig. 2d). Correlating the relative time participants spent sampling the cooperate row and their cooperation rate showed that participants indeed cooperated more often when they looked longer at the cooperate row, but only when this row was placed at the bottom (Fig. 6b, right panel, correlation between the relative time spent sampling the cooperate row and cooperation rates; $r(86) = 0.49$, $p < 0.001$, CI = [0.31 0.63] for Cooperate-down, $r(86) = 0.1$, $p = 0.3$, CI = [−0.11 0.31] for Cooperate-up). A linear regression of the effects of the relative sampling time of the cooperate row, row position, and their interaction on cooperation rate confirmed that the interaction between relative sampling time and row position was significant ($\beta = 1.28$, $p = 0.0023$, CI = [0.47 2.11]—note that the same relations were found between the proportion of trials starting with fixation to the cooperate row and participant's cooperation rates; see Supplementary Fig. 8). One possibility is that this

decreased sampling of the temptation payoff T for the participants, but it is unlikely that attention to this payoff drives changes in cooperation rates (Fig. 4a).

This pattern of results may reflect individual differences in participants' goal-directed intentions or capacity to overcome a first-gaze bias. Indeed, sampling the cooperation row when it is placed on top is part of the habitual first gaze (Fig. 3a–c), but this does not strongly influence cooperation. Sampling cooperation payoffs in the bottom row, on the other hand, is not habitual. Participants who overcome the pattern and sample the cooperation AOIs despite their location have a higher cooperation rate, suggesting that the participant-specific capacity to overcome a first-gaze-to-the-top-row bias is linked to cooperation. In line with this interpretation, the difference in cooperation between the row positions was strongly correlated with the differences in the relative sampling time of the cooperate row between positions (Fig. 6c, right panel, correlation between differences in relative sampling time and cooperation rates between the two-row positions, $r(86) = 0.67$, $p < 0.001$, CI = [0.54 0.77]). The option position Cooperate-down-left—which has the highest cooperation rate—combines both the effect of cooperation placed left on sampling other's information, and the effect of overcoming attentional bias on sampling cooperation information. This may explain the particularly high cooperation rates when participants gazed long at this position.

Finally, as Fig. 2d and Supplementary Table 1 (column 2) show, a change in row order (the opponent's choice options) affects cooperation rates. As Fig. 6a shows, the change in row order primarily affects the time spent looking at the cooperation row; this time is linked to cooperation rates (Fig. 6b, c). This suggests that the effect of specific option positions on cooperation may be mediated by how a specific option position attracts attention.

To confirm this mediation effect between these separate observations, we performed the following analyses: In the mixed-effects regression model showing the effect of the row order on cooperation, we included an effect that is likely responsible for the change in cooperation rate: the relative duration of sampling the cooperation row. This regression analysis shows that the change in row order indeed affects the duration of gaze at the cooperation row (Supplementary Table 10, column 1, $\beta = -0.1$, $p < 0.001$, CI = [−0.13 −0.07]) and that this gaze duration in turn is linked to more cooperative choices (Supplementary Table 10, column 2, $\beta = 0.53$, $p < 0.001$, CI = [0.33 0.73]).

A final mediation test confirmed that the effect of the row order on cooperative choice is driven by the indirect mediation effects of the gaze variables (Supplementary Fig. 9, $\beta = -0.02$, CI = [−0.03 −0.01], $p < 0.001$). While gaze by itself is a significant predictor of cooperation, it does not fully explain the variation in cooperative choices, and the row order remains a significant predictor of cooperation (with the row order and gaze on cooperation row contributing with opposite signs).

## Discussion

Our results demonstrate that cooperative behavior is not only affected by the potential outcomes of social interactions and individual preferences but also by attentional mechanisms (at least when outcomes are viewed on a computer screen as here). Attention to others' payoffs was associated with increased cooperation rates, at both the individual and game levels. Exogenous manipulation of the payoff presentation, independent of the manipulation of payoff values, changed participants' gaze patterns, leading to changes in cooperation rates. Participants' attention at the beginning of a trial was naturally attracted to the top-left part of the screen and to higher payoffs. Changes in the option presentation therefore affected which payoff information was sampled by the participants, leading them to specific sampling sequences (importantly, increasing the chance to sample payoffs of the opponent). Sequences of fixations to the different payoffs, combined with the payoff values, predicted trial-by-trial cooperation rates, showing the importance of information sampling patterns in cooperative decisions.

Our study is the first to systematically evaluate cooperation decisions using a wide range of payoffs within the same participant, allowing us to

**a.** Relative time spent sampling the payoffs of Other or the Cooperate row in the four option positions

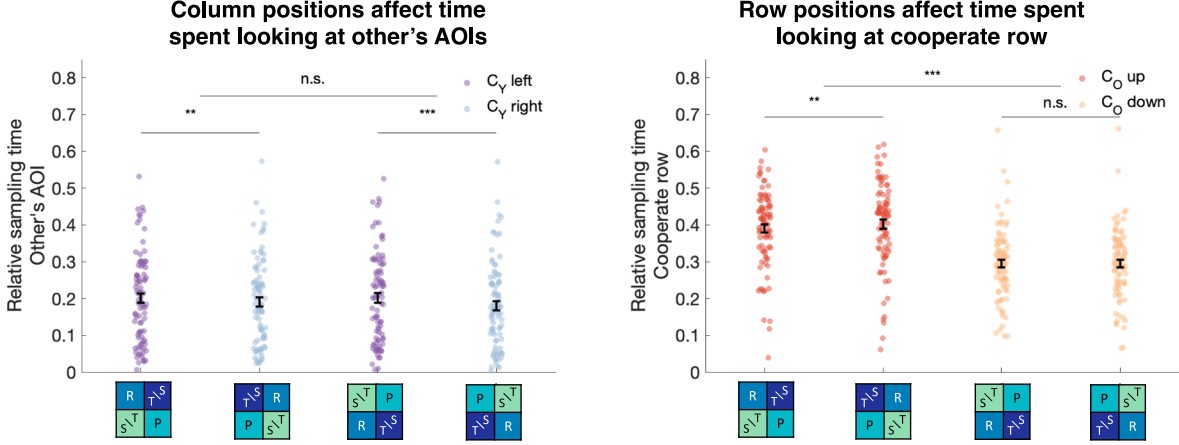

**b.** Relative time spent sampling the payoffs of Other or the Cooperate row affects cooperation rates

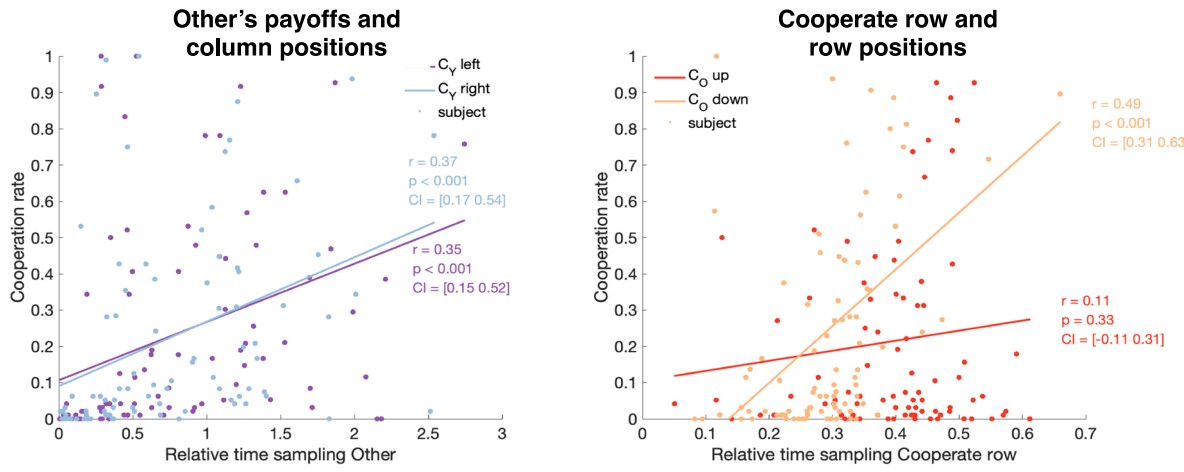

**c.** Differences in relative time spent sampling the payoffs the Cooperate row correlate with individual differences in cooperation rates

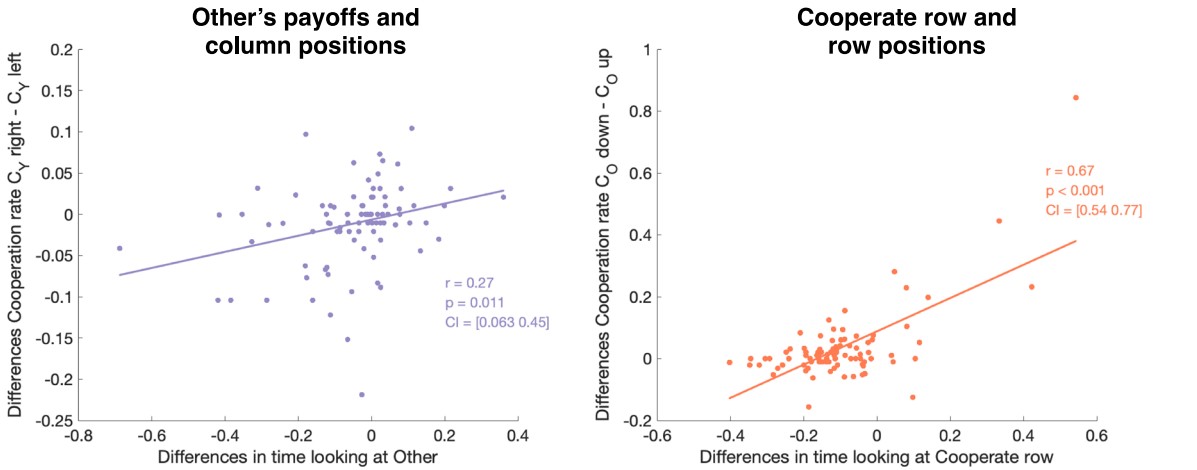

simultaneously estimate beliefs, risk, and other-regarding preferences using one-shot Prisoner's Dilemma games. Previous studies have shown that cooperation in such games was influenced by the different payoff values, with simplified indices derived from payoffs used to predict cooperation across games and showing the importance of different motives in cooperation decisions[9–15]. Here we used a different approach, explaining

individual rather than group behavior using utility models incorporating standard economic choice model assumptions such as risk preferences (stemming from diminishing marginal utility)[61], other-regarding preferences[16], and expected utility calculation using prior beliefs. Using an extensive set of models, we found that individuals' behavior was best captured by a combination of other-regarding preferences and beliefs about the

**Fig. 6 | Effects of relative spent sampling payoffs of Other or the cooperate row on cooperation and interactions with options positions. a** Relative time spent sampling Other's AOIs (left) or AOIs on the cooperate row (right) for different column and row positions. The relative times are computed for each participant (represented with a dot per participant), the error bars represent the average across participants and s.e.m. Significance stars represent a two-tailed paired *t*-test: $p < 0.00033$ (***), $p < 0.0033$ (**), $p < 0.017$ (*), or non-significant (n.s.), using a Bonferroni correction. In the left panel, the colors represent the column positions: Cooperate-left vs Defect-left. In the right panel, the colors represent the row positions: Cooperate-up vs Defect-up. **b** Correlations between the relative time spent sampling other's AOIs (left panel) or the cooperate row (right panel) and the cooperation rate of each participant, for the different positions of the columns (left panel) or rows (right panel). The dots represent individual participants and the lines and *p*-values show the results of regressions between the two variables across participants. **c** Differences in time spent sampling others' AOI (left panel) or the cooperate row (right panel) between the two column positions (left panel) or row positions (right panel) and differences in cooperation rates between the positions. The dots represent individual participants and the lines and *p*-values show the results of regressions between the two variables across participants. See Supplementary Fig. 8 for an analogous analysis restricted to the first fixation. All panels: $N = 88$ participants.

opponent's choices, while risk preferences (on average) did not play an important role. In our study, however, outcomes were deterministic and only depended on both participants' choices. Future studies could use stochastic outcomes to evaluate the link between risk preferences and cooperative behavior in settings that are closer to real-world decisions.

Previous studies have linked cooperative behavior to information sampling patterns[37,38], showing that gaze directed at others' payoffs is typically associated with cooperation. These studies, focusing on goal-directed (endogenous) attentional mechanisms, highlight individual differences in cooperation that can be associated with different sampling strategies, reflecting individuals' levels of pro-sociality[38], rationality (playing according to game-theoretic predictions), and strategic sophistication[37]. Supporting these findings, we found that participants who attended to the opponent's payoffs—and in particular the opponents' reward – cooperated more, while participants attending primarily to their payoffs often defected. These effects may reflect that people with a greater ability to mentalize and employ theory-of-mind[63,64] may focus attention on others' outcomes to determine whether to cooperate. In addition to these participant-level effects, we found that within-trial gaze sequences that included the opponent's payoff were also associated with an increased cooperation rate for those trials. To disentangle the informational and preferential components of these attentional effects, future studies could use an asymmetric payoff matrix that orthogonally manipulates rewards for the participant and their opponent, creating an additional incentive to sample own or other's payoffs.

We found that exogenous manipulations affecting bottom-up attention impacted cooperation. Simple permutations of option positions on the screen affected individual cooperation rates, independently from choice outcomes or individual preferences. These results provide insights for the important debate on the question of whether attention (affected for instance by gaze manipulation) can change the way preferences are constructed or whether it purely affects the choice process itself[47,65]. Previous studies focusing on the effects of goal-directed attention have shown an effect of attention on the subjective values attributed to attended options[26,65,66], which may partly explain behaviors such as preference reversal[67–69] or instability[70].

Studies exogenously driving attention to some choice options in non-social domains found that such manipulations can bias choices[47,71,72]. Extending such exogenous manipulations to social decisions, we found that attention affected information sampling and choices, but not estimated social preferences. These findings suggest that even though our display manipulations changed attention towards the other's payoffs, they did not seem to change the perception of inequality (captured by the social preference parameter of the model) but instead appeared to increase prosocial tendencies overall (captured by the cooperation belief parameter). These findings suggest that new models may be needed that explicitly address the interplay between these factors that influence subjective valuation processes underlying prosocial behavior. It is unclear whether the mediation effect we found was specifically driven by a change in expectations of cooperation, as captured by model parameter *p*. Given that the model is estimated at the participant level and *p* was not a significant predictor of the individual cooperation rate, we cannot reliably confirm this link with our data.

Examining potential causes of bottom-up attentional effects, we found that a habitual information sampling pattern[37,55], attributed to the left-to-right and up-to-down Western writing conventions[56,57], may partially underlie these effects. Simple alterations of choice interfaces, such as

changing the position of the options presented therefore influence information sampling and choices[53,73,74] by interacting with this natural pattern. Neural evidence suggests that initial information processing plays an important role in social decisions[75]. Here we find that manipulating the position of the options on the screen changed cooperation rates but did not significantly increase the selection of the top-left option. Instead, after a first fixation on higher payoffs placed on the upper part of the screen, more complex mechanisms may have oriented participants' attention toward specific gaze sequences.

Previous studies of binary choices largely focused on the effects of the first and last fixations as well as the relative dwell time of each option on decisions across multiple domains[26,47,76]. Cooperation decisions in the Prisoner's Dilemma, however, are more complex, as four different payoffs are represented for two different participants, and the outcomes are conditional on others' decisions. Recent studies of multi-attribute decisions in non-social decisions have extended findings that the first fixation and the relative dwell time of different options are related to choices[77], and have shown that gaze allocations to options and attributes have interactive influences on choices[46].

Studies on multi-attribute decisions in social dilemmas have focused on the frequencies of transitions between fixations of different payoffs and have found that these reflected distinct social preferences in dictator games[36,38] or different degrees of strategic sophistication in strategic games[37]. However, these studies neither evaluated the temporal structure of information sampling nor the potential differential effects of fixations occurring in different temporal order. Our results showed that the first and last fixations or overall dwell times were not sufficient to accurately predict cooperation and that options sampled second had a high effect on behavior.

Going further than this overall dwell time, our analysis highlighted the importance of specific sampling sequences and showed that the order of the fixations on different payoffs was key to predicting cooperation. Using machine learning, we were able to accurately predict trial-by-trial cooperation from five-fixation (four first and last) gaze sequences, with higher accuracy than using the first and last fixations only, showing the importance of studying gaze sequences. Scanpaths have been used in vision research to compare information-sampling patterns across individuals[78] or to improve users' experience in human-computer interactions[79], and they have recently been applied to predict choices in economic games[80].

Here we showed that studying such paths in the context of cooperative decisions improves predictions of behavior and allows us to explain the effects of exogenous manipulations of attention on choices. Positioning options at different locations on the screen modified the full gaze patterns with which participants sampled information, not just the first information participants looked at. Extending our results to different games would allow us to test whether the gaze patterns we measured here generalize across games[37] and individuals.

## Limitations

Modifying gaze behavior may affect the content of the information participants consider for their decision, or the way this information is processed. More work is needed to uncover the specific computational mechanisms of individual differences in gaze strategies and how exactly they influence choices. Further studies could manipulate the temporal presentation of options[49], or the size and saliency[81] of the different payoffs

and response options, to evaluate whether these manipulations could further increase cooperation. Testing our protocol with populations of non-western origin with different reading conventions, such as right-to-left Hebrew or Arabic, or top-to-bottom Chinese or Japanese readers could additionally provide more insight into how different habitual reading patterns combined with manipulations of information presentation may affect social choices.

Finally, our experimental setup contained a simple manipulation of option presentation in a Prisoner's Dilemma that may only partially mimick information sampling in real-life situations. Nevertheless, our results demonstrate that social behavior can be exogenously influenced toward more pro-social choices via clearly measurable attentional mechanisms. Previous studies have suggested the general importance of choice architecture for nudging behavioral outcomes[82–84]. Our findings provide a more mechanistic, attention-based account that could have an impact on the design of real-world interventions, such as healthcare measures. One speculative example could be new public health policies, such as printed information materials drawing individuals' attention toward the specific outcomes of their decisions for others, which may lead to an increase in cooperation. Another potential application is in the charitable giving domain, where it has been shown that drawing attention to specific aspects of donations can increase charitable behavior[85–87]. Future empirical studies are needed to test the application of our findings in these real-world interventions.

## Data availability
The data sets generated during and analyzed for the current study are publicly available in the OSF repository at: https://osf.io/z5pb7/ (https://doi.org/10.17605/OSF.IO/Z5PB7).

## Code availability
The code reproducing the analysis is publicly available in the OSF depository at: https://osf.io/z5pb7/. The study used R 4.2.2, R packages lme4 1.1.18.1, lmerTest 3.0.1, Matlab 2020a.

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

## Acknowledgements
This project has received funding from the European Research Council (ERC) under the European Union's Horizon 2020 research and innovation program (grant agreement no. 725355, ERC consolidator grant BRAINCODES). C.L. was funded by the Marlene Porsche Graduate School in Neuroeconomics and C.C.R. received funding from the University Research Priority Program 'Adaptive Brain Circuits in Development and Learning' (grant no. URPP AdaBD) at the University of Zurich and the Swiss National Science Foundation (grant no. 100019L-173248). The funders had no role in study design, data collection and analysis, decision to publish or preparation of the manuscript. Open Access costs were financed by the University Library Zurich.

## Author contributions
All authors designed the experiment and analyses. C.L. and A.K. programmed the experiment. C.L. and A.K. conducted the experiment. C.L. and A.K. performed the data analysis. All authors co-wrote the paper. C.C.R. supervised the project.

## Competing interests
The authors declare no competing interests.
