## [Transparent Peer Review file · Communications Psychology]

Manipulating Attention Facilitates Cooperation

Corresponding Author: Dr Arkady Konovalov

Version 0:

Decision Letter: first round

Dear Professor Konovalov,

Thank you for your patience during the peer-review process. Your manuscript titled "Facilitating Cooperation by Manipulating Attention" has now been seen by 3 reviewers, and I include their comments at the end of this message. They find your work of interest but raised some important points. We are interested in the possibility of publishing your study in Communications Psychology, but would like to consider your responses to these concerns and assess a revised manuscript before we make a final decision on publication.

We therefore invite you to revise and resubmit your manuscript, along with a point-by-point response to the reviewers. Please highlight all changes in the manuscript text file.

Editorially, we consider important that your revision incorporate the additional control analyses requested by the reviewers, notably regarding the trial number and normalized payoffs, as well as the parameter recovery analysis and confusion matrix for the different model candidates. Your revision should also follow the reviewers' recommendations for additional analyses aiming at elucidating the mechanisms driving the effects (e.g., mediation analyses for attention patterns and models with multiple parameters representing prior beliefs). Finally, following some of the reviewers' comments, we would also encourage you to consider the idea to perform a confirmatory pre-registered study, or in its absence, to at least further discuss the exploratory nature of your findings.

I am attaching an Editorial Requests Table that details critical reporting requirements for the revised manuscript. Please attend to each item and ensure your manuscript is fully compliant. We are requesting that your manuscript aligns with these requirements as this facilitates the evaluation of your manuscript, reducing delays in re-review and potential future acceptance. If your revised manuscript is not aligned with these requests on major issues, such as those concerning statistics, it may be returned to you for further revisions without re-review. Additional information can be found in our style and formatting guide <https://www.nature.com/documents/commpsychol-style-formatting-guide-accept.pdf> Communications Psychology formatting guide.

Please use the following link to submit your

- revised manuscript,
- point-by-point response to the referees' comments,
- cover letter (as a separate document),
- the Editorial Policy Checklist (see below),
- the Reporting Summary (see below), and
- the completed Editorial Request Table (attached):

Link Redacted

** This url links to your confidential home page and associated information about manuscripts you may have submitted or be

reviewing for us. If you wish to forward this email to co-authors, please delete the link to your homepage first **

Best regards,

Mael Lebreton

Mael Lebreton, PhD
Editorial Board Member
Communications Psychology
orcid.org/0000-0002-2071-4890

REVIEWER EXPERTISE:

Reviewer #1 computational modelling; strategic decision-making, behavioural game theory

Reviewer #2 attention & preferences

Reviewer #3 computational modelling; strategic decision-making, behavioural game theory

REVIEWER REPORTS:

Reviewer #1 (Remarks to the Author):

This paper examines the role of attention in cooperative behavior. The participants played a prisoner's dilemma (PD) while their gaze was recorded using an eye-tracking device. The authors varied the position of the different payoffs of the game across trials and estimated a social preferences utility model to assess the determinant of the participants' cooperative behavior. Participants' behavior and model's parameters relate to their attentional pattern and the manipulation of the position of the different payoffs affects their level of cooperation. Overall, I found the topic of the paper timely and interesting. It contributes the large literature on attention and social behavior. The analysis are well executed and show a clear understanding of the literature although I found the paper a bit dense and sometimes hard to follow due to the numerous analysis. Yet, I have several comments I would like the authors to address.

My first main comment is about the repetitions of the PD. I acknowledge that the lack of feedback between each trial makes it "as if" it was a one shot game, however the potential dynamics across the trials are never controlled in the analysis (boredom, exploration etc...). The authors need to add a trial number control variable in every regression to ensure that their different results are robust to those dynamics.

My second main comment is about the lack of a parameter recovery analysis of the main model. Indeed, as some results are based on the parameters from a computational choice model I think this is important to test whether the parameter of this model is stable. On a similar note, a confusion matrix for the different models candidates (Supp Table 8) would be also beneficial to show the robustness of the model selection procedure.

Minor comments:

- The size of the payoffs is a potential confound for the regression reported in Supplementary Table 2. The relative sampling time can be influenced by the size of the rectangle representing the different payoffs. I think adding the normalized payoffs as control variables would make the results more robust.
- A note in the Supplementary Table 3 stating why there is only 66 subjects in the regression (instead of 88) would be beneficial.

Reviewer #2 (Remarks to the Author):

Summary: This study explores the role of attention in cooperation decisions in independent prisoner's dilemmas. Behaviorally, the authors find that cooperation is linked to a model parameter representing other-regarding preferences. In line with previous findings on goal-directed attention, they find that those who look more at other's payoffs are more cooperative and that the other-regarding parameter correlates with dwell time on other's payoffs.

The paper's key contribution to the literature is in manipulating stimulus-driven attention by changing the position of options (cooperate or defect for self and other) across trials within-participants. This enables the authors to investigate how this exogenous attention manipulation impacts choices and attentional patterns. Behaviorally, they find more cooperation when the partner's cooperation option is on bottom, especially if the participant's own cooperation option is on the left (mutual cooperation on the bottom left). They look at attentional patterns related to cooperation and differences in attentional patterns caused by the manipulation to elucidate why positioning mutual cooperation on bottom facilitates cooperation. Linking attention to behavior independently of the manipulation, they find that first fixations are most often to top and left AOIs and participants fixate more on their own payoffs on top and the partner's payoffs on the left. Moreover, fixating first to one's own payoffs is related to reduced cooperation, and fixating first on other's payoffs is related to more cooperation, which does not fully explain the manipulation findings. Therefore, they explore more complex gaze sequences and show that models predict cooperation best if they include the first four and last fixations, and specifically find evidence that looking at other's cooperation reward as the second fixation is linked to higher cooperation. At the same time, the authors show that option positions impact which combinations of AOIs are viewed in the first 2 fixations, suggestive evidence that some option positions may lead to more cooperation by fostering 2nd fixations to mutual cooperation. Finally, the authors link differences in first fixations and dwell time depending on option positions to differences in cooperation. They find that the cooperation option on the left is linked to more attention to other's payoffs and more cooperation. In terms of row position, they find that attention to the cooperation row is only linked to cooperative behavior when the cooperation row is on bottom, suggesting that only when it is sought out does it impact the decision.

Evaluation: The question of how exogenous, stimulus-driven attention impacts cooperation is interesting, and the findings presented are relevant to researchers trying to link prosocial behavior and attention. The authors' finding that presenting the mutual cooperation option on bottom leads to more cooperation is somewhat surprising since one might assume the top-left reading bias of their sample would mutual cooperation on the top-left would lead to the highest cooperation. The authors try to unpack this counterintuitive finding by exploring more complex attentional patterns related to cooperation and investigating how option positions impact attentional patterns, showing that gaze patterns beyond the first fixation and overall dwell time matter for understanding behavior. That said, the paper is quite complicated with many analyses, making it hard to follow at points and hard to focus in on the most important attentional mechanisms. In particular, some attentional patterns are linked to the option position manipulation and gaze patterns are linked to behavior, but the bridge between is vague. Further, the lack of preregistration along with the exploration of many attentional patterns means that replication would make the case much stronger or at least more acknowledgement of the exploratory nature, points which I discuss more below.

Main comments: A lot of work clearly went into understanding the counterintuitive results of the manipulation (assuming that the expectation was that mutual cooperation in the top left AOI would lead to more cooperation), and I do think the authors convincingly show that attention metrics beyond first fixation and dwell time matter for understanding cooperative behavior more generally. I appreciate the transparency in showing all of the analyses that went into trying to make sense of the manipulation; however, the volume of analyses of complex gaze patterns also has implications for structuring and presenting the findings:

1. The paper does not mention hypotheses or preregistration, so my understanding is that it was exploratory and data-driven. Especially because the findings are not as expected and involve many different analyses, it is not clear how robust all of the findings are. A follow-up experiment replicating the main findings would be ideal. If doing a confirmatory study, it would probably cleaner to remove the bars representing magnitude to not also manipulate stimulus-driven attention in a second way since this does not seem to be a main focus of the findings. I realize a confirmatory study is a large ask, so at least a more explicit acknowledgement of the exploratory nature of these findings and highlighting those that are most important to test in future studies would be helpful.
2. The authors find interesting and potentially relevant patterns in 1) how gaze sequences impact cooperation behavior and 2) how the option manipulation affects patterns of gaze but only relate them broadly in the text, making it unclear whether these are indeed the attentional mechanisms through which the option manipulation impacts behavior. For example, a pattern that comes out most prominently is 1) that looking at other's cooperation reward as the second fixation is linked to more cooperation behavior and 2) certain option positions are associated with a higher frequency of looking at other's reward from cooperation as the second fixation. However, these two findings are only suggestively linked in the text, but there is no direct evidence that this is the channel through which information position impacts cooperation. Therefore, quantifying how much of the position effect is explained by the attention patterns would be very helpful (e.g., via mediation or another method) as many interesting potential mechanisms are discussed without a clear picture of which may be the most important. Relatedly, more of a link to theory or hypotheses for future research about why such patterns were found would help contextualize these findings.
3. Similarly, because there are so many different patterns shown, the authors tend to only comment on the most striking findings, which seems more anecdotal and makes it hard to discern the main takeaways. For example, there are analyses of how the first 4 fixations relate to cooperation as well as the most common two-fixation dyads and 4-fixations sequences for each option position, but only some of the patterns are commented on. Given that 2nd fixation on other's cooperation reward seems to come out of multiple analyses, I wonder if more focus on this as a main channel rather than all of the potential patterns would help narrow it down to the most promising mechanisms with other analyses in the supplement? In addition to second fixations to other's reward, Figure 6 (and Supp. Fig. 6) which focus on dwell time and first fixation show the clearest link between the position manipulation, attention, and cooperation, so I would present both of these in the main text.
4. There is a lot of discussion about the impact of placing the mutual cooperation AOI on the bottom row, but I also wonder if some of the position effects on increasing cooperation could be due to the fact that when mutual cooperation is on the bottom row, so is the participant's defection payoff, making it less salient? Perhaps Figure 4a (top left panel)/Supp. Table 7 provide evidence against this interpretation, but from a theoretical perspective, it seems that making one's own highest temptation to defect less salient could also reduce defection.

5. The authors find that presenting mutual cooperation on the bottom leads to a higher parameter representing beliefs about the partner's cooperation. However, this is not definitively linked to specific attentional patterns and in the individual difference analyses this parameter is not related to individual cooperation rates. Therefore, a bit more on the interpretation/discussion of the meaning of this difference would be nice.
6. Since participants play many rounds, were there any order effects and did these interact with the position manipulation?

Minor points:

7. Fig. 1, panel B "cooperate" is not showing up in the correct place in the horizontal/YOU axis.
8. The figure legends are very detailed. Information that is present in the text, especially interpretation, does not need to be in the legends (e.g., the full description of the prisoners dilemma games), and the same goes for information that is already described within the figure (e.g., U = upper row and L = left column in Fig. 3).
9. In Supp. Table 2, stars supposedly both indicate significance as does bolding—what is the difference, and do both need to be there?
10. On p. 6, regressions of relative sampling times for each AOI on cooperation are referenced, but the corresponding table number is not provided. Are these regressions or simply the correlations presented in the Fig. 2c?
11. At the bottom of p. 7, the description of analyses is a bit convoluted with many nested parentheses. The explanation of what was done becomes clearer when looking at Supp. Fig. 2b-d. Since this finding rules out other effects, for ease of reading, I might take this out of the main text and simply show it in the supplement. Regardless, I would emphasize the intuitive interpretation which is that people who vary on the different parameters might be more or less sensitive to the attention manipulation (e.g., those who are more other-regarding could be more impacted by it) but this is not the case. Relatedly, it would be easier to interpret Supp. Fig. 2b-d if you kept the y-axis consistent so that the 0 positions lined up.
12. The Fig. 3 legend references Fig. 4e to find the average frequency of first fixations for different AOIs, but Fig. 4e doesn't exist. I assume it is meant to refer to Fig. 3c?
13. In the supplementary tables, the organization of variables seems to change partway through the tables at times, making comparisons difficult (e.g., Supp. Table 6 in the first column orders by AOI, then you/other and last fixation order whereas the second column orders by fixation order with alternating T-you and S-other).
14. At the bottom of p.12, the authors refer to Supp. Fig. 4 for frequent sequence details, but I think this should be Supp. Fig. 5?

Reviewer #3 (Remarks to the Author):

In the manuscript entitled "Facilitating Cooperation by Manipulating Attention" the authors address a very interesting question about how attention influences cooperation. Specifically, the authors investigate the role of exogenous manipulation of attention in influencing decisions to be cooperative in a one-shot prisoner's dilemma (PD), using a combination of computational modelling and eye-tracking.

They use a range of different PD games to model individual other-regarding preferences, risk attitudes, and subjective prior beliefs about potential cooperation by the co-player and manipulate attention by changing the size of the rectangles containing information about the different payoff values, while varying the position of the payoffs across trials.

They found the model best describing the data was one where subjects used their prior beliefs about the likely cooperation of the co-player, to compute an utility that took into account their risk aversion and their other regarding preferences.

They found highly significant correlation between gaze preferences for the co-player payoffs and cooperation rates and other regarding preferences estimated by the model. They also find mild effects of the position of payoffs on the screen on choices and on the parameter representing beliefs about cooperation. These effects were further mediated by the gaze sequence which affected cooperation rates and could be used through machine learning classification to predict cooperative choices.

Overall, this manuscript provides a clear, well written and comprehensive analysis of the effect of attention on cooperation in prisoner's dilemmas. Its findings on the effects of exogenous manipulation of bottom up attention on cooperation are interesting and convincing and are novel and worthy of attention in the field. I commend the clarity of the manuscript and the thoroughness of the analysis.

I have only a number of relatively minor comments that could hopefully help the authors improve the manuscript further.

Comments

It is not that surprising that the risk aversion parameter does not correlate with choices as it seems like the prior belief about cooperation should already capture the inherent social risk of the choice and the social risk is the only risk here as the outcomes are deterministic. This point should be better highlighted in the discussion and throughout the paper. This is consistent with the fact that the alternative models without the risk aversion parameter or with it fixed to a certain value have statistically similar fits.

A related question stems: does the prior belief parameter correlate with choice in the model that assumes risk-neutrality? And if not, why not? One would imagine that an expectation of cooperation would be a major factor in determining choices to cooperate as reciprocity/tiffortat is an important driver of cooperation.

Another related question: it is somewhat confusing the fact that the only parameter that is affected by the exogenous manipulation of attention is the prior belief parameter which appears not to be correlated with choices. It seems therefore unlikely that attention affects cooperation rates by boosting the expectation of reciprocity. How do the authors interpret this and what is their suggested mechanism for how attention affects cooperation rates, given that other regarding preferences parameters are unaffected (fig.2E)?

What seems to be the most intuitive explanation of the lack of effect of the prior belief is that beliefs about others' cooperation will not be fixed but will vary both as a function of the co-player identity and the payoff distributions. Given that the subjects did not know the co-player, the only source of variability across trials would be the payoff distributions. Given that parameters are fit across all trials, they can only capture an average expectation which is not modulated by payoffs. But while it is reasonable to assume that other regarding preferences might be fixed across trials for each participant as they express their general pro-sociality, expectations of reciprocity are very likely to change in a similar way as cooperation rates depend on payoffs. To check whether that is the case the authors could run another model where multiple parameters representing prior beliefs in different prisoners' dilemma could capture some of this variability. Using a median split for the difference between reward and punishment could provide two blocks of PD games to test this. If there is, how one could expect, a higher prior belief of cooperation in trials where R is much bigger than P. Such results would be consistent with the view that expectations of cooperation are strongly modulated across PD games and could be the cognitive mechanism which is affected by the information sampling strategy.

The finding that the cooperation row position strongly affects behaviour is interesting as well as the fact that the time spent (and the number of first fixation) on the top row do not affect cooperation rates but they do on the bottom row. The authors interpret this as the ability to overcome a first-gaze bias. But another explanation could be that irrespective of the first-gaze bias, any gaze successive to the first which is most often top left, will be more goal directed and therefore expression of an intention. In fact the authors report very modest first gaze biases judging from figure 4a. Could the authors better explain the supposed effect? And could also confirm whether an opposite effect holds for the time spent looking at the defect row depending on whether it is top or bottom?

Minor points

Was there a relationship across subjects between the belief parameter and the other regarding preferences?

The paragraph "To identify these bottom-up effects, we systematically manipulated the position of the different options on the screen. On each trial, we randomly permuted the positions of the options on the screen" avoid the repetition.

What was the range of alpha values? Can the authors add plots for the parameters distribution across subjects? The y axis in figure 3b seems not to be percentages. Please adjust.

EDITORIAL POLICIES

We ask that you ensure your manuscript complies with our editorial policies and reporting requirements.

To that end, we require revised manuscripts to be accompanied by two completed items: a reporting summary that collects information on study design and procedure, and an editorial policy checklist that verifies compliance with all required editorial policies.

- <https://www.nature.com/documents/nr-reporting-summary.zip> Nature Research Reporting Summary
- <https://www.nature.com/documents/nr-editorial-policy-checklist.pdf> Editorial Policy Checklist

All points on the policy checklist must be addressed. Your revised manuscript can only be sent back to the referees if these checklists are completed and uploaded with the revision.

Notes: If you have submitted a Stage 1 Registered Report, Review, Primer, Comment, or Perspective you do not need to submit these forms. If you have already submitted these forms, you may disregard this request.

If you experience problems in linking your ORCID, please contact the Platform Support Helpdesk.

Version 1: second round

Decision Letter:

Dear Professor Konovalov,

Your manuscript titled "Facilitating Cooperation by Manipulating Attention" has now been seen by our reviewers, whose comments appear below. In light of their advice I am delighted to say that we are happy, in principle, to publish a suitably revised version in Communications Psychology.

We therefore invite you to revise your paper one last time to address the remaining concerns of our reviewers and a list of editorial requests. At the same time we ask that you edit your manuscript to comply with our format requirements and to maximise the accessibility and therefore the impact of your work.

EDITORIAL REQUESTS:

SUBMISSION INFORMATION:

OPEN ACCESS:

* TRANSPARENT PEER REVIEW: Communications Psychology uses a transparent peer review system. On author request, confidential information and data can be removed from the published reviewer reports and rebuttal letters prior to publication. If you are concerned about the release of confidential data, please let us know specifically what information you would like to have removed. Please note that we cannot incorporate redactions for any other reasons.

* CODE AVAILABILITY: All Communications Psychology manuscripts must include a section titled "Code Availability" at the

end of the methods section. We require that the custom analysis code supporting your conclusions is made available in a publicly accessible repository at this stage; please choose a repository that generates a digital object identifier (DOI) for the code; the link to the repository and the DOI must be included in the Code Availability statement. Publication as Supplementary Information will not suffice.

* DATA AVAILABILITY:

Link Redacted

Best regards,

Jennifer Bellingtier

Jennifer Bellingtier, PhD
Senior Editor
Communications Psychology

Mael Lebreton, PhD
Editorial Board Member
Communications Psychology
orcid.org/0000-0002-2071-4890

REVIEWERS' EXPERTISE:

Reviewer #1 computational modelling; strategic decision-making, behavioural game theory
Reviewer #2 attention & preferences
Reviewer #3 computational modelling; strategic decision-making, behavioural game theory

REVIEWERS' COMMENTS:

Reviewer #1 (Remarks to the Author):

The authors have addressed all of my concerns with the original manuscript.

Reviewer #2 (Remarks to the Author):

My concerns have been addressed, and I appreciate the effort by the authors put into to showing the link between the attention manipulation, attentional patterns, and choice.

Very minor comments:

- In the new description of the temporal analyses for serial effects there is redundancy on p. 9 "found no influence no significant effects", so "no influence" can be removed.
- The phrasing of the regression significance in the last sentence of the first paragraph on p.15 is confusing, "the main effect of AOI located left, $p < 0.001$ for all AOIs other" and could perhaps be clarified as "for all AOIs of other's payoffs."

Reviewer #3 (See attachment)

RESPONSE TO REFEREES

Reviewer #1 (Remarks to the Author):

Overall, I found the topic of the paper timely and interesting. It contributes the large literature on attention and social behavior. The analysis are well executed and show a clear understanding of the literature although I found the paper a bit dense and sometimes hard to follow due to the numerous analysis. Yet, I have several comments I would like the authors to address.

We thank the reviewer for this positive evaluation of our contribution and the thoughtful feedback!

My first main comment is about the repetitions of the PD. I acknowledge that the lack of feedback between each trial makes it “as if” it was a one shot game, however the potential dynamics across the trials are never controlled in the analysis (boredom, exploration etc...). The authors need to add a trial number control variable in every regression to ensure that their different results are robust to those dynamics.

This is a fair point - even though many other studies have taken a similar approach to ours in repeating one-shot PD games. We now confirm the stability of our choices across time in two ways. The first of them is the consistency of choices (both at the subject and the game level) between the first and the second blocks of the experiment (which contain the same set of games). We now report these statistics in the main text:

“Cooperation rates were stable between the first and second parts of the experiment, both on the subject level (Pearson correlation $r = 0.9$, $t(64) = 15.9$) and the game level (Pearson correlation $r = 0.72$, $t(94) = 10.05$).”

Additionally, we followed the reviewer’s suggestion and added a trial number control variable in all regression analyses in the paper. The results largely remain unchanged with this control. Specifically, Supplementary Tables 1, 2, 6, and 7, which included observations on the individual trial level, have been updated to include the trial number as a control variable. We hope the reviewer agrees that these analyses show that choices were stable across time and that potential dynamics across time are no concern.

My second main comment is about the lack of a parameter recovery analysis of the main model. Indeed, as some results are based on the parameters from

a computational choice model I think this is important to test whether the parameter of this model is stable. On a similar note, a confusion matrix for the different models candidates (Supp Table 8) would be also beneficial to show the robustness of the model selection procedure.

We apologize for this omission. We did perform the parameter recovery exercise but did not include these results in the manuscript, as we assumed such a check is a default for model fitting. We have now added the parameter recovery matrix to the supplements, which shows that the parameters can be recovered well (Figure R1).

Figure R1. Pearson correlations between the simulated and recovered parameters using the full model; 1000 datasets were simulated using random parameter sets in ranges $\beta = (0, 1.3)$; $\alpha = (0, 1)$; $\rho = (1, 1.3)$; $p = (0, 1)$ and recovered using the MLE procedure.

Given that the models we are testing are very similar in nature (some of the models are simply degenerate versions of the winning model, and all models are variants of the same expected utility model with slight variation in the form of the social and risk preference functions), a confusion matrix analysis did not yield a clear distinction between the models (e.g., a model with a smaller number of parameters often, on average, across many uniformly drawn values of parameters, leads in the BIC comparison if the benefit of variability in a fixed parameter is rather small). We recognize that this is a drawback of our model comparison process, but we believe it does not invalidate it.

Minor comments:

- **The size of the payoffs is a potential confound for the regression reported in Supplementary Table 2. The relative sampling time can be influenced by the size of the rectangle representing the different payoffs. I think adding the normalized payoffs as control variables would make the results more robust.**

Good point – we added the normalized payoffs as a control in this regression. Supplementary Table 2 now contains updated values. Note that adding this variable does not change the results qualitatively: While the payoffs themselves are significant predictors of trial-by-trial cooperation, the relative sampling times still provide a significant explanation of variance in cooperation (as confirmed by many other analyses in the paper, including out of sample machine learning predictions). Thus, payoff size is not a confound for our reported effects.

- **A note in the Supplementary Table 3 stating why there is only 66 subjects in the regression (instead of 88) would be beneficial.**

We have added a note to the table explaining this discrepancy. In short, this regression only included the subjects that had a computational model fit (had more than one cooperative or non-cooperative choice trial, so the model could not provide a reliable parameter fit). We have now made sure the reader understands this.

Thanks to this reviewer for the insightful comments, which have helped us to strengthen our paper!

Reviewer #2 (Remarks to the Author):

Evaluation: The question of how exogenous, stimulus-driven attention impacts cooperation is interesting, and the findings presented are relevant to researchers trying to link prosocial behavior and attention. The authors' finding that presenting the mutual cooperation option on bottom leads to more cooperation is somewhat surprising since one might assume the top-left reading bias of their sample would mutual cooperation on the top-left would lead to the highest cooperation. The authors try to unpack this counterintuitive finding by exploring more complex attentional patterns related to cooperation and investigating how option positions impact attentional patterns, showing that gaze patterns beyond the first fixation and overall dwell time matter for understanding behavior.

We thank the reviewer for this positive evaluation of our work and for the thoughtful feedback!

That said, the paper is quite complicated with many analyses, making it hard to follow at points and hard to focus in on the most important attentional mechanisms. In particular, some attentional patterns are linked to the option position manipulation and gaze patterns are linked to behavior, but the bridge between is vague. Further, the lack of preregistration along with the exploration of many attentional patterns means that replication would make the case much stronger or at least more acknowledgement of the exploratory nature, points which I discuss more below.

We appreciate this honest feedback. It is true that we included many analyses, which may complicate the overall flow. However, these analyses were necessary to unpack the complex mechanisms at work in the behavior captured by our detailed data. We acknowledge the point about missing preregistration but note that our study was intentionally exploratory, since we did not have detailed hypotheses about specific relationships between attention and cooperation. We thank the reviewer for pointing us to the ambiguity in that respect and have now added a clear indication of this to the manuscript:

We had no prior hypotheses on the complex relationship between the cooperative choices, gaze data, and our display manipulation, so we clearly indicate that our results are primarily exploratory. However, we used pilot data to confirm that individuals do respond to trial-by-trial variation in payoffs. Additionally, we replicated the behavioral analyses using the companion fMRI study (Supplementary Table 1, part b).

1. The paper does not mention hypotheses or preregistration, so my understanding is that it was exploratory and data-driven. Especially because the findings are not as expected and involve many different analyses, it is not clear how robust all of the findings are. A follow-up experiment replicating the main findings would be ideal. If doing a confirmatory study, it would probably be cleaner to remove the bars representing magnitude to not also manipulate stimulus-driven attention in a second way since this does not seem to be a main focus of the findings. I realize a confirmatory study is a large ask, so at least a more explicit acknowledgement of the exploratory nature of these findings and highlighting those that are most important to test in future studies would be helpful.

As mentioned above, our study was intentionally exploratory. Unfortunately, given the timeline and funding of the project, the current employment of the first authors, and time constraints for the revision, we are unable to run a confirmatory pre-registered eye-tracking study. However, we have made sure that the exploratory nature of our study is very clear in the revised manuscript (as was intended from the start, since we did not have a priori hypotheses about the specific relationships between payoff positions, gaze data, and cooperative choices).

2. The authors find interesting and potentially relevant patterns in 1) how gaze sequences impact cooperation behavior and 2) how the option manipulation affects patterns of gaze but only relate them broadly in the text, making it unclear whether these are indeed the attentional mechanisms through which the option manipulation impacts behavior. For example, a pattern that comes out most prominently is 1) that looking at other's cooperation reward as the second fixation is linked to more cooperation behavior and 2) certain option positions are associated with a higher frequency of looking at other's reward from cooperation as the second fixation. However, these two findings are only suggestively linked in the text, but there is no direct evidence that this is the channel through which information position impacts cooperation. Therefore, quantifying how much of the position effect is explained by the attention patterns would be very helpful (e.g., via mediation or another method) as many interesting potential mechanisms are discussed without a clear picture of which may be the most important. Relatedly, more of a link to theory or hypotheses for future research about why such patterns were found would help contextualize these findings.

We agree that these effects are among the study's most interesting observations and that a mediation analysis will strengthen our findings. We have now added a new analysis along the lines suggested by the reviewer that - we believe - clearly demonstrates that option positions affect the cooperation rates via attentional mechanisms.

As Figure 2d and Supplementary Table 1 (column 2) show, a change in row order (the opponent's choice options) affects cooperation rates. Supplementary Table 2 and Figure 2c demonstrate that longer sampling of the other's payoff is associated with higher cooperation rates. Finally, Supplementary Table 7 shows specifically that fixation on the other's R payoff is linked to higher cooperation rates.

To confirm the mediation effect between these separate observations, we performed the following analyses: In the regression model showing the effect of the row order on cooperation, we included the two effects that are thought to be responsible for the identified effects: Either the total duration of sampling of the other's payoffs or a dummy variable indicating that the second fixation was to the other's Reward payoff. In both of these analyses, including the mechanism variable led to the effect of the order becoming insignificant, indicating that this effect is primarily mediated by the gaze variables. To confirm this, we ran two mediation analyses. These both underline that row order affects cooperation through the gaze variables.

Figure R2. (a) Mediation analysis results using individual trial choice (= 1 if cooperate) as the dependent variable, the trial-level row order as the independent variable, and relative gaze to other's payoffs as a mediator. (b) Mediation analysis identical to panel (a), using the dummy variable indicating that the second fixation of the trial was to the other's Reward payoff. All regressions are mixed effects models including trial number as a control variable and subject-level random effects.

Both tests showed that the effect of the row order on cooperative choice is driven by the indirect mediation effects (Figure R2, $p = 0.01$ for the total sampling time for the other's payoffs and $p < 0.001$ for the dummy indicator of the second fixation being the other's Reward payoff). We have added these analyses to the text, which now provides a clearer

picture of the relation between the different results and the mechanisms by which row order influences cooperation.

In addition to these analyses, we have added passages to the Discussion on how attention can influence cooperation, as well as on implications for future research. This provides the theoretical embedding suggested by the reviewer.

In addition to these subject-level effects, we found that within-trial gaze sequences that included the opponent's payoff were also associated with an increased cooperation rate for those trials. To disentangle the informational and preferential components of these attentional effects, future studies could use an asymmetric payoff matrix that orthogonally manipulates rewards for the subject and their opponent, creating an additional incentive to sample own or other's payoffs.

[...]

These findings suggest that even though our display manipulations changed attention towards the other's payoffs, they did not seem to change the perception of inequality (captured by the social preference parameter of the model) but instead appeared to increase prosocial tendencies overall (captured by the cooperation belief parameter). These findings suggest that new models may be needed that explicitly address the interplay between these factors that influence subjective valuation processes underlying prosocial behavior.

3. Similarly, because there are so many different patterns shown, the authors tend to only comment on the most striking findings, which seems more anecdotal and makes it hard to discern the main takeaways. For example, there are analyses of how the first 4 fixations relate to cooperation as well as the most common two-fixation dyads and 4-fixations sequences for each option position, but only some of the patterns are commented on. Given that 2nd fixation on other's cooperation reward seems to come out of multiple analyses, I wonder if more focus on this as a main channel rather than all of the potential patterns would help narrow it down to the most promising mechanisms with other analyses in the supplement? In addition to second fixations to other's reward, Figure 6 (and Supp. Fig. 6) which focus on dwell time and first fixation show the clearest link between the position manipulation, attention, and cooperation, so I would present both of these in the main text.

Given the exploratory nature of the study, we tried to provide a comprehensive picture of the data rather than singling out some specific results. However, with the benefit of hindsight, we agree that it makes sense to focus on the most promising mechanisms. As

we showed above, our data suggest that attention to the other's payoffs increases with changes in the option positions and thus drives increased cooperation. We have edited the text to highlight this potential mechanism more clearly. Specifically, we tried to remove some unnecessary details, analyses, and repetitions. We hope the flow of the argument is now improved.

Given that Supplementary Figure 6 is simply a version of Figure 6 that uses first fixation rather than total duration time, and that reports a very similar set of results (perhaps this was unclear in our presentation), we are hesitant to present both analyses in the main text. However, to prevent related misunderstandings in future readers, we have clarified the discussion of both figures in the text.

4. There is a lot of discussion about the impact of placing the mutual cooperation AOI on the bottom row, but I also wonder if some of the position effects on increasing cooperation could be due to the fact that when mutual cooperation is on the bottom row, so is the participant's defection payoff, making it less salient? Perhaps Figure 4a (top left panel)/Supp. Table 7 provide evidence against this interpretation, but from a theoretical perspective, it seems that making one's own highest temptation to defect less salient could also reduce defection.

This is an interesting suggestion. We have included a mention of this possibility in the text, but as the reviewer points out, attention to this payoff does not drive changes in cooperation (as Figure 4a clearly shows).

5. The authors find that presenting mutual cooperation on the bottom leads to a higher parameter representing beliefs about the partner's cooperation. However, this is not definitively linked to specific attentional patterns and in the individual difference analyses this parameter is not related to individual cooperation rates. Therefore, a bit more on the interpretation/discussion of the meaning of this difference would be nice.

Thank you for pointing out this omission. We see the difference between the social preference parameter α and the belief p as follows: While α reflects aversion to inequality (and thus ensures differential response to different payoff combinations), the belief parameter p simply captures the overall tendency to cooperate across all payoff combinations. Given the design of the experiment, it can capture several potential motivations (e.g. reciprocity expectations or general prosociality). In the new discussion,

we explain this point and state that more elaborate experiments are needed to pin down these distinctions.

6. Since participants play many rounds, were there any order effects and did these interact with the position manipulation?

It is natural for this task to have an autocorrelation in choices (subjects who are likely to cooperate on one trial are also typically more likely to cooperate on the following trial). However, we found no interaction effect with row order ($p = 0.65$). We also found no significant effects of the previous trial's payoffs (R, P, T, S) on the current trial's choice. We added these observations to the results.

Minor points:

7. Fig. 1, panel B “cooperate” is not showing up in the correct place in the horizontal/YOU axis.

Thank you, fixed!

8. The figure legends are very detailed. Information that is present in the text, especially interpretation, does not need to be in the legends (e.g., the full description of the prisoners dilemma games), and the same goes for information that is already described within the figure (e.g., U = upper row and L = left column in Fig. 3).

We have reduced the amount of text in the figure legends, with the most significant changes in the caption of Figure 1. The captions now only present the information needed to understand the plots.

9. In Supp. Table 2, stars supposedly both indicate significance as does bolding—what is the difference, and do both need to be there?

In this and other tables, we intended to use bolding to draw attention to the specific coefficients discussed in the text. Perhaps this is just confusing, so we removed the bolding indication from all tables.

10. On p. 6, regressions of relative sampling times for each AOI on cooperation are referenced, but the corresponding table number is not provided. Are these regressions or simply the correlations presented in the Fig. 2c?

This was a typo; it should be “correlations”, not “regressions”. These are indeed the correlations in Figure 2c. We corrected the sentence.

11. At the bottom of p. 7, the description of analyses is a bit convoluted with many nested parentheses. The explanation of what was done becomes clearer when looking at Supp. Fig. 2b-d. Since this finding rules out other effects, for ease of reading, I might take this out of the main text and simply show it in the supplement. Regardless, I would emphasize the intuitive interpretation which is that people who vary on the different parameters might be more or less sensitive to the attention manipulation (e.g., those who are more other-regarding could be more impacted by it) but this is not the case. Relatedly, it would be easier to interpret Supp. Fig. 2b-d if you kept the y-axis consistent so that the 0 positions lined up.

Thank you for these suggestions. We simplified the text here and adjusted the supplementary figures as suggested.

12. The Fig. 3 legend references Fig. 4e to find the average frequency of first fixations for different AOIs, but Fig. 4e doesn't exist. I assume it is meant to refer to Fig. 3c?

Correct, this should have referred to Figure 3c. Fixed now!

13. In the supplementary tables, the organization of variables seems to change partway through the tables at times, making comparisons difficult (e.g., Supp. Table 6 in the first column orders by AOI, then you/other and last fixation order whereas the second column orders by fixation order with alternating T-you and S-other).

We have adjusted the organization of the variables to simplify the comparisons. The payoffs are now ordered the same way as in the figures in all tables (first Temptation, then Reward, Punishment, Suckers).

14. At the bottom of p.12, the authors refer to Supp. Fig. 4 for frequent sequence details, but I think this should be Supp. Fig. 5?

Correct, this is fixed now.

Thanks also to this reviewer for the thoughtful comments and suggestions that have helped to streamline our result presentation.

Reviewer #3 (Remarks to the Author):

Overall, this manuscript provides a clear, well written and comprehensive analysis of the effect of attention on cooperation in prisoner's dilemmas. Its findings on the effects of exogenous manipulation of bottom up attention on cooperation are interesting and convincing and are novel and worthy of attention in the field. I commend the clarity of the manuscript and the thoroughness of the analysis.

We thank the reviewer for the positive evaluation of our paper and for their thoughtful comments.

It is not that surprising that the risk aversion parameter does not correlate with choices as it seems like the prior belief about cooperation should already capture the inherent social risk of the choice and the social risk is the only risk here as the outcomes are deterministic. This point should be better highlighted in the discussion and throughout the paper. This is consistent with the fact that the alternative models without the risk aversion parameter or with it fixed to a certain value have statistically similar fits.

We used the term “risk” here primarily to indicate the potential curvature of the utility function as typically used in the economic literature (so higher payoffs have diminishing

Figure R3. Simulation of the choice probabilities in the main model using $p = 0.5$, $\beta = 0.2$, and variation in α from 0 to 1 and the risk parameter ρ at values 0.7, 1, and 1.3.

utility to the decision maker, which leads to risk aversion given that the outcome of the other's choice is not deterministic and can be viewed as a lottery given the prior belief). There might be several reasons why we do not see that individual variability in risk-

aversion does not improve the model fits. However, these comments inspired us to investigate this question more deeply.

Simulations of the model using the task demonstrate that there is no linear relationship between the risk parameter and cooperation if social preferences also vary. Specifically, altruistic individuals decrease their cooperation in the task with higher levels of risk aversion, while in selfish individuals cooperation increases with increased risk aversion (see Figure R3 above).

Thus, even though we did not find a direct correlation between the risk aversion parameter and cooperation ($r(86) = 0.01$, $p = 0.94$), we ran a new regression analysis at the subject level, including all parameters as predictors of individual cooperation rates. When controlling for all model parameters, we found that risk aversion also explains a significant part of the variance in cooperation rates (R^2 without ρ : 0.75, R^2 with ρ : 0.81; coefficient for ρ in the full model: $\beta = -0.35$, $t = 4.4$, $p < 0.001$). However, these effects are small and thus associated with very small differences in model fits between the full model and the model with varying risk attitudes. We have now included this result and a discussion of it in the paper.

To be transparent here, we expected a larger impact of risk perception on choices when we started this project, but the data point to a fairly small role, which we now report.

A related question stems: does the prior belief parameter correlate with choice in the model that assumes risk-neutrality? And if not, why not? One would imagine that an expectation of cooperation would be a major factor in determining choices to cooperate as reciprocity/titfortat is an important driver of cooperation.

As in the main model, there was no significant simple correlation between the prior belief parameter and the cooperation rate on the subject level in the model that assumed risk neutrality ($r = 0.16$, $p = 0.2$). However, when we ran a regression including all model parameters, we found a significant effect of the prior belief parameter ($\beta = 0.16$, $t = 2.4$, $p = 0.02$). This confirms the idea that the expectation of cooperation does have an effect if one assumes risk neutrality; however, the effect of the prior belief is not significant in the full model, so these two parameters may indeed share some variance. We noted this result in the paper.

Another related question: it is somewhat confusing the fact that the only parameter that is affected by the exogenous manipulation of attention is the prior belief parameter which appears not to be correlated with choices. It

seems therefore unlikely that attention affects cooperation rates by boosting the expectation of reciprocity. How do the authors interpret this and what is their suggested mechanism for how attention affects cooperation rates, given that other regarding preferences parameters are unaffected (fig.2E)?

As we showed above, given the non-linear nature of the model, we do not expect a simple correlation between the belief parameter and the cooperative choices on the individual level. Furthermore, the belief parameter reflects the general tendency to cooperate across all trials and can capture several potential mechanisms, including reciprocity and attentional drivers.

We have now included additional analyses showing that change in the row order affects cooperative choices through attention to the other's payoffs (Figure R4).

Figure R4. (a) Mediation analysis results using individual trial choice (= 1 if cooperate) as the dependent variable, the trial-level row order as the independent variable, and relative gaze to other's payoffs as a mediator. (b) Mediation analysis identical to panel (a), using the dummy variable indicating that the second fixation of the trial was to the other's Reward payoff. All regressions are mixed effects models including trial number as a control variable and subject-level random effects.

We included these analyses in the paper and tried to clarify the central point of our findings.

What seems to be the most intuitive explanation of the lack of effect of the prior belief is that beliefs about others' cooperation will not be fixed but will vary both as a function of the co-player identity and the payoff distributions. Given that the subjects did not know the co-player, the only source of variability across trials would be the payoff distributions. Given that parameters are fit across all trials, they can only capture an average expectation which is not modulated by payoffs. But while it is reasonable to assume that other regarding preferences might be fixed across trials for

each participant as they express their general pro-sociality, expectations of reciprocity are very likely to change in a similar way as cooperation rates depend on payoffs. To check whether that is the case the authors could run another model where multiple parameters representing prior beliefs in different prisoners' dilemma could capture some of this variability. Using a median split for the difference between reward and punishment could provide two blocks of PD games to test this. If there is, how one could expect, a higher prior belief of cooperation in trials where R is much bigger than P. Such results would be consistent with the view that expectations of cooperation are strongly modulated across PD games and could be the cognitive mechanism which is affected by the information sampling strategy.

This is an interesting suggestion. As suggested by the reviewer, we fit a new model, using our best-fitting model as the baseline, but assuming that the prior belief parameter p varies across two types of trials (using a median split on R – P difference, which is 10 CHF). While we found a small increase in p for trials with higher R-P difference (0.47 vs 0.43 across all subjects), this increase was not statistically significant ($p = 0.54$). Given that there was no significant effect, we chose not to include this analysis in the paper, but would be happy to do so if necessary.

The finding that the cooperation row position strongly affects behaviour is interesting as well as the fact that the time spent (and the number of first fixation) on the top row do not affect cooperation rates but they do on the bottom row. The authors interpret this as the ability to overcome a first-gaze bias. But another explanation could be that irrespective of the first-gaze bias, any gaze successive to the first which is most often top left, will be more goal directed and therefore expression of an intention. In fact the authors report very modest first gaze biases judging from figure 4a. Could the authors better explain the supposed effect? And could also confirm whether an opposite effect holds for the time spent looking at the defect row depending on whether it is top or bottom?

We need to point out that we used the *relative, not the absolute*, time sampling of the Cooperate row (relative to the Defect row), so the “opposite” effect mentioned by the reviewer naturally holds.

To clarify our interpretation: cooperative individuals tend to look around more (in general and including the other's payoffs) than non-cooperative ones. Given that the gaze typically starts at the top row (first-gaze bias), it is natural that these individuals will be

more likely to look at the bottom row (to locate the R payoff), and the looking time for that row would be correlated with cooperation rates.

We agree that goal-direct gaze interpretation is also viable, however potentially it should be stronger for the columns as the intention is to choose one of the columns, not rows (if we interpret the idea of the “expression of an intention” correctly). In general, more cooperative subjects should also be more goal-directed, so we believe this interpretation aligns with our suggestions. We adjusted the text to reflect this idea.

Minor points

Was there a relationship across subjects between the belief parameter and the other regarding preferences?

There was no significant relationship between these two parameters (see Figure R5 below). We added this observation to the text.

Figure R5. Scatterplot of the identified social preference (α) and belief (p). Top left corner: Pearson correlation $r = -0.18$.

We now also included a parameter recovery exercise showing there is no correlation between those parameters in the simulations as well.

The paragraph “To identify these bottom-up effects, we systematically manipulated the position of the different options on the screen. On each trial, we randomly permuted the positions of the options on the screen” avoid the repetition.

Fixed:

To identify these bottom-up effects, we randomly permuted the positions of the options on the screen on each trial, in a within-subject and counterbalanced manner, independently from the payoff values presented.

What was the range of alpha values? Can the authors add plots for the parameters distribution across subjects?

We have now added the parameter distribution plots to the supplements:

Figure R6. Parameter distribution plots for the winning model.

The y axis in figure 3b seems not to be percentages. Please adjust.

Fixed!

Thanks for these suggestions, which have helped us to improve the manuscript!

Reviewer 2

Very minor comments:

- In the new description of the temporal analyses for serial effects there is redundancy on p. 9 “found no influence no significant effects”, so “no influence” can be removed.

Fixed!

- The phrasing of the regression significance in the last sentence of the first paragraph on p.15 is confusing, “the main effect of AOI located left, $p < 0.001$ for all AOIs other” and could perhaps be clarified as “for all AOIs of other's payoffs.”

Fixed!

Reviewer 3

I am a bit confused by the figure which seems to have the axis flipped (altruistic individuals would have $\alpha = 1$ and high cooperation not the opposite) and what the authors refer to by saying “in the task with higher levels of risk aversion”. As I understand, the risk parameter is fit to each subject, not task.

We apologize for the flipped axis and the poor wording here. The full sentence is “individuals decrease their cooperation in the task with higher levels of risk aversion,” and it is supposed to mean “individuals who have higher levels of risk aversion would decrease their cooperation in this task.” So, indeed, the parameter reflects the subject's risk aversion. We have now changed the figure and this sentence to avoid this ambiguity.

Assuming the axis is indeed flipped, what it shows is that the risk parameter would have opposite effects on cooperation rates based on social preference, which is consistent with the fact that the risk parameter captures the diminishing utility of higher payoffs for both the subject payoffs and those of their co-player and the shift in focus between the two can flip the effect. This is interesting and whilst orthogonal to the main results reported about gaze could be at least mentioned and referred to in the supplementary material.

Thank you for this suggestion. We have fixed this plot ($\alpha = 1$ now reflects higher degree of social preference) and have added it, accompanied by commentary, to the supplementary materials (Supplementary Figure 9).

However I am still not convinced the risk parameter captures the risk implied by the task, nor that it would do so independently of the prior belief

parameter. I wonder whether the measure of game riskiness defined in reference 59, which is indeed task related, can be used to predict cooperation rates trial to trial as it would be independent of the subject fit prior belief parameter.

Regarding whether the risk parameter captures the risk implied by the task, our parameter recovery exercise (Supplementary Figure 9a) shows that, in principle, this parameter can be reliably recovered independently of the prior belief parameter. It just does not seem to play a role in this specific dataset given the model fits. We now further emphasize this point in the text on p.7 to avoid that this question comes up in future readers.

The measure of game riskiness, as defined by reference 59, is also known as the cooperation index developed by Rapport and Chammah (reference 13) and is well-known to predict cooperation rates. It is a compound index that uses all 4 payoffs. It does not necessarily reflect risk per se and is not based on theoretical considerations about individual preferences (as it does not incorporate any parameters). Given the large literature on the effects of payoffs and recent work on this that we cite in references 9-15, we believe that these effects are outside of the scope of our paper, which focuses on individual preferences and traits.

These mediation analyses are convincing but could the author expand on whether they think the effect on the prior belief parameter reported in fig2D suggests that a second fixation to other's R boost the expectation of cooperation?

We could not identify a clear link here given our data. It is unclear whether this effect is driven by a change in expectations of cooperation, as captured by model parameter p . The model is estimated at the subject level, and p was not a significant predictor of the individual cooperation rate, so the underlying relationship might be more complex than a simple correlation between gaze data and the belief parameter. We have added the following passage in the Discussion section to make this explicit:

It is unclear whether the mediation effect we found was specifically driven by a change in expectations of cooperation, as captured by model parameter p . Given that the model is estimated at the participant level and p was not a significant predictor of the individual cooperation rate, we cannot reliably confirm this link with our data.

I think it would be useful to discuss this result, albeit just a trend and not significant, in conjunction with the result of figure 2D, suggesting a boost in the expectation of cooperation as the psychological route to boosting cooperation based on other payoffs' fixations. It could help qualify the "prosocial tendencies" mentioned in the added paragraph in the discussion.

We added the suggested model to the model comparison (Supplementary Tables 2 and 3) and have added the following comment:

While we found a small increase in p in trials with a higher R—P difference (0.47 vs 0.43 across all participants), this increase was not statistically significant ($t(65) = 0.65$, $p = 0.52$).

You may want to check the consistency of the labels on the axis.

Thank you, we double-checked these!

In the manuscript entitled “Facilitating Cooperation by Manipulating Attention” the authors address a very interesting question about how attention influences cooperation. Specifically, the authors investigate the role of exogenous manipulation of attention in influencing decisions to be cooperative in a one-shot prisoner’s dilemma (PD), using a combination of computational modelling and eye-tracking.

They use a range of different PD games to model individual other-regarding preferences, risk attitudes, and subjective prior beliefs about potential cooperation by the co-player and manipulate attention by changing the size of the rectangles containing information about the different payoff values, while varying the position of the payoffs across trials.

They found the model best describing the data was one where subjects used their prior beliefs about the likely cooperation of the co-player, to compute an utility that took into account their risk aversion and their other regarding preferences.

They found highly significant correlation between gaze preferences for the co-player payoffs and cooperation rates and other regarding preferences estimated by the model. They also find mild effects of the position of payoffs on the screen on choices and on the parameter representing beliefs about cooperation. These effects were further mediated by the gaze sequence which affected cooperation rates and could be used through machine learning classification to predict cooperative choices.

Overall, this manuscript provides a clear, well written and comprehensive analysis of the effect of attention on cooperation in prisoner’s dilemmas. Its findings on the effects of exogenous manipulation of bottom up attention on cooperation are interesting and convincing and are novel and worthy of attention in the field. I commend the clarity of the manuscript and the thoroughness of the analysis.

I have only a number of relatively minor comments that could hopefully help the authors improve the manuscript further.

Comments

It is not that surprising that the risk aversion parameter does not correlate with choices as it seems like the prior belief about cooperation should already capture the inherent social risk of the choice and the social risk is the only risk here as the outcomes are deterministic. This point should be better highlighted in the discussion and throughout the paper. This is consistent with the fact that the alternative models without the risk aversion parameter or with it fixed to a certain value have statistically similar fits.

We used the term “risk” here primarily to indicate the potential curvature of the utility function as typically used in the economic literature (so higher payoffs have diminishing

Figure R3. Simulation of the choice probabilities in the main model using $p = 0.5$, $\beta = 0.2$, and variation in α from 0 to 1 and the risk parameter ρ at values 0.7, 1, and 1.3.

utility to the decision maker, which leads to risk aversion given that the outcome of the other's choice is not deterministic and can be viewed as a lottery given the prior belief). There might be several reasons why we do not see that individual variability in risk aversion does not improve the model fits. However, these comments inspired us to investigate this question more deeply.

Simulations of the model using the task demonstrate that there is no linear relationship between the risk parameter and cooperation if social preferences also vary. Specifically, altruistic individuals decrease their cooperation in the task with higher levels of risk aversion, while in selfish individuals cooperation increases with increased risk aversion (see Figure R3 above).

Thus, even though we did not find a direct correlation between the risk aversion parameter and cooperation ($r(86) = 0.01$, $p = 0.94$), we ran a new regression analysis at the subject level, including all parameters as predictors of individual cooperation rates. When controlling for all model parameters, we found that risk aversion also explains a significant part of the variance in cooperation rates (R^2 without ρ : 0.75, R^2 with ρ : 0.81; coefficient for ρ in the full model: $\beta = -0.35$, $t = 4.4$, $p < 0.001$). However, these effects are small and thus

associated with very small differences in model fits between the full model and the model with varying risk attitudes. We have now included this result and a discussion of it in the paper.

To be transparent here, we expected a larger impact of risk perception on choices when we started this project, but the data point to a fairly small role, which we now report.

I am a bit confused by the figure which seems to have the axis flipped (altruistic individuals would have $\alpha = 1$ and high cooperation not the opposite) and what the authors refer to by saying "in the task with higher levels of risk aversion". As I understand, the risk parameter is fit to each subject, not task.

Assuming the axis is indeed flipped, what it shows is that the risk parameter would have opposite effects on cooperation rates based on social preference, which is consistent with the fact that the risk parameter captures the diminishing utility of higher payoffs for both the subject payoffs and those of their co-player and the shift in focus between the two can flip the effect. This is interesting and whilst orthogonal to the main results reported about gaze could be at least mentioned and referred to in the supplementary material.

However I am still not convinced the risk parameter captures the risk implied by the task, nor that it would do so independently of the prior belief parameter. I wonder whether the measure of game riskiness defined in reference 59, which is indeed task related, can be used to predict cooperation rates trial to trial as it would be independent of the subject fit prior belief parameter.

A related question stems: does the prior belief parameter correlate with choice in the model that assumes risk-neutrality? And if not, why not? One would imagine that an expectation of cooperation would be a major factor in determining choices to cooperate as reciprocity/titfortat is an important driver of cooperation.

As in the main model, there was no significant simple correlation between the prior belief parameter and the cooperation rate on the subject level in the model that assumed risk neutrality ($r = 0.16$, $p = 0.2$). However, when we ran a regression including all model parameters, we found a significant effect of the prior belief parameter ($\beta = 0.16$, $t = 2.4$, $p = 0.02$). This confirms the idea that the expectation of cooperation does have an effect if one assumes risk neutrality; however, the effect of the prior belief is not significant in the full model, so these two parameters may indeed share some variance. We noted this result in the paper.

Another related question: it is somewhat confusing the fact that the only parameter that is affected by the exogenous manipulation of attention is the prior belief parameter which appears not to be correlated with choices. It seems therefore unlikely that attention affects cooperation rates by boosting the expectation of reciprocity. How do the authors interpret this and what is their suggested mechanism for how attention affects cooperation rates, given that other regarding preferences parameters are unaffected (fig.2E)?

As we showed above, given the non-linear nature of the model, we do not expect a simple correlation between the belief parameter and the cooperative choices on the individual level.

Furthermore, the belief parameter reflects the general tendency to cooperate across all trials and can capture several potential mechanisms, including reciprocity and attentional drivers. We have now included additional analyses showing that change in the row order affects cooperative choices through attention to the other's payoffs (Figure R4).

Figure R4. (a) Mediation analysis results using individual trial choice (= 1 if cooperate) as the dependent variable, the trial-level row order as the independent variable, and relative gaze to other's payoffs as a mediator. (b) Mediation analysis identical to panel (a), using the dummy variable indicating that the second fixation of the trial was to the other's Reward payoff. All regressions are mixed effects models including trial number as a control variable and subject-level random effects.

We included these analyses in the paper and tried to clarify the central point of our findings.

These mediation analyses are convincing but could the author expand on whether they think the effect on the prior belief parameter reported in fig2D suggests that a second fixation to other's R boost the expectation of cooperation?

What seems to be the most intuitive explanation of the lack of effect of the prior belief is that beliefs about others' cooperation will not be fixed but will vary both as a function of the co-player identity and the payoff distributions. Given that the subjects did not know the co-player, the only source of variability across trials would be the payoff distributions. Given that parameters are fit across all trials, they can only capture an average expectation which is not modulated by payoffs. But while it is reasonable to assume that other regarding preferences might be fixed across trials for each participant as they express their general pro-sociality, expectations of reciprocity are very likely to change in a similar way as cooperation rates depend on payoffs. To check whether that is the case the authors could run another model where multiple parameters representing prior beliefs in different prisoners' dilemma could capture some of this variability. Using a median split for the difference between reward and punishment could provide two blocks of PD games to test this. If there is, how one could expect, a higher prior belief of cooperation in trials where R is much bigger than P. Such results would be consistent with the view that expectations of cooperation are strongly modulated across PD games and could be the cognitive mechanism which is affected by the information sampling strategy.

This is an interesting suggestion. As suggested by the reviewer, we fit a new model, using our best-fitting model as the baseline, but assuming that the prior belief parameter p varies across two types of trials (using a median split on $R - P$ difference, which is 10 CHF). While we found a small increase in p for trials with higher $R-P$ difference (0.47 vs 0.43 across all subjects), this increase was not statistically significant ($p = 0.54$). Given that there was no

significant effect, we chose not to include this analysis in the paper, but would be happy to do so if necessary.

I think it would be useful to discuss this result, albeit just a trend and not significant, in conjunction with the result of figure 2D, suggesting a boost in the expectation of cooperation as the psychological route to boosting cooperation based on other payoffs' fixations. It could help qualify the "prosocial tendencies" mentioned in the added paragraph in the discussion.

The finding that the cooperation row position strongly affects behaviour is interesting as well as the fact that the time spent (and the number of first fixation) on the top row do not affect cooperation rates but they do on the bottom row. The authors interpret this as the ability to overcome a first-gaze bias. But another explanation could be that irrespective of the first-gaze bias, any gaze successive to the first which is most often top left, will be more goal directed and therefore expression of an intention. In fact the authors report very modest first gaze biases judging from figure 4a. Could the authors better explain the supposed effect? And could also confirm whether an opposite effect holds for the time spent looking at the defect row depending on whether it is top or bottom?

We need to point out that we used the relative, not the absolute, time sampling of the Cooperate row (relative to the Defect row), so the "opposite" effect mentioned by the reviewer naturally holds. To clarify our interpretation: cooperative individuals tend to look around more (in general and including the other's payoffs) than non-cooperative ones. Given that the gaze typically starts at the top row (first-gaze bias), it is natural that these individuals will be 15 more likely to look at the bottom row (to locate the R payoff), and the looking time for that row would be correlated with cooperation rates. We agree that goal-direct gaze interpretation is also viable, however potentially it should be stronger for the columns as the intention is to choose one of the columns, not rows (if we interpret the idea of the "expression of an intention" correctly). In general, more cooperative subjects should also be more goal-directed, so we believe this interpretation aligns with our suggestions. We adjusted the text to reflect this idea.

Minor points

Was there a relationship across subjects between the belief parameter and the other regarding preferences?

There was no significant relationship between these two parameters (see Figure R5 below). We added this observation to the text.

Figure R5. Scatterplot of the identified social preference (α) and belief (p). Top left corner: Pearson correlation $r = -0.18$.

We now also included a parameter recovery exercise showing there is no correlation between those parameters in the simulations as well.

The paragraph “To identify these bottom-up effects, we systematically manipulated the position of the different options on the screen. On each trial, we randomly permuted the positions of the options on the screen” avoid the repetition.

Fixed: To identify these bottom-up effects, we randomly permuted the positions of the options on the screen on each trial, in a within-subject and counterbalanced manner, independently from the payoff values presented.

What was the range of alpha values? Can the authors add plots for the parameters distribution across subjects?

We have now added the parameter distribution plots to the supplements:

The y axis in figure 3b seems not to be percentages. Please adjust.

Fixed!

Thanks for these suggestions, which have helped us to improve the manuscript

You may want to check the consistency of the labels on the axis.